# Optimal Control for Transformer Architectures: Enhancing Generalization, Robustness and Efficiency

**Kelvin Kan**
Department of Mathematics
UCLA
kelvin.kan@math.ucla.edu

**Xingjian Li**
Oden Institute
University of Texas at Austin
xingjian.li@austin.utexas.edu

**Benjamin J. Zhang**
School of Data Science and Society
UNC Chapel Hill
bjz@unc.edu

**Tuhin Sahai**
SRI International
tuhin.sahai@sri.com

**Stanley Osher**
Department of Mathematics
UCLA
sjo@math.ucla.edu

**Markos A. Katsoulakis**
Department of Mathematics and Statistics
University of Massachusetts Amherst
markos@umass.edu

## Abstract

We study Transformers through the perspective of optimal control theory, using tools from continuous-time formulations to derive actionable insights into training and architecture design. This framework improves the performance of existing Transformer models while providing desirable theoretical guarantees, including generalization and robustness. Our framework is designed to be plug-and-play, enabling seamless integration with established Transformer models and requiring only slight changes to the implementation. We conduct seven extensive experiments on tasks motivated by text generation, sentiment analysis, image classification, and point cloud classification. Experimental results show that the framework improves the test performance of the baselines, while being more parameter-efficient. On character-level text generation with `nanoGPT`, our framework achieves a 46% reduction in final test loss while using 42% fewer parameters. On `GPT-2`, our framework achieves a 9.3% reduction in final test loss, demonstrating scalability to larger models. To the best of our knowledge, this is the first work that applies optimal control theory to both the training and architecture of Transformers. It offers a new foundation for systematic, theory-driven improvements and moves beyond costly trial-and-error approaches.

## 1   Introduction

Transformers have achieved state-of-the-art performance in various applications, including natural language processing [69, 83], computer vision [22], program synthesis [14], computational biology [43], speech processing [4], reinforcement learning [13, 57], operator learning [56, 96] and climate modeling [31, 60, 61].

The popularity of Transformers has led to myriad architectural variants [17, 19, 87], each developed to exhibit certain advantages. Practitioners, however, often discover effective Transformer architectures through a trial-and-error approach. The objective function used to train Transformers admits a natural

39th Conference on Neural Information Processing Systems (NeurIPS 2025).

formulation as an optimal control problem, with the loss serving as a terminal cost and model depth corresponding to time. By examining the optimality conditions of this formulation, optimal transport (OT) emerges as a principled regularizer that ensures well-posedness. This *optimal control framework* informs both the structure of the loss — through regularization — and the design of the final layer, grounding these architectural choices in control-theoretic principles rather than heuristics.

Guided by this framework, we propose OT-Transformer, a *plug-and-play* model grounded in optimal control theory. Our model is flexible and straightforward to implement in the sense that one can directly insert a predefined Transformer architecture into the OT-Transformer model. This requires only slight modifications to existing code and allows seamless integration with established models. An illustration of our framework is given in Figure 1.

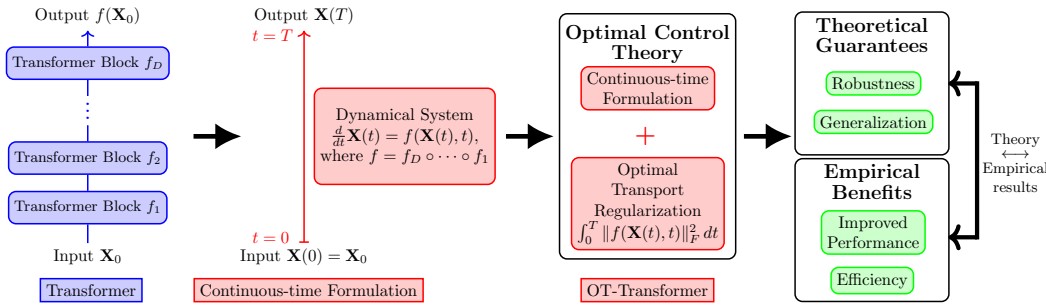

Figure 1: **Schematic Illustration of our Plug-and-Play Model** (8). Given an existing Transformer, we construct a continuous-time formulation by using the Transformer to parametrize the velocity field of a dynamical system. Optimal control theory informs the learning of the velocity field by imposing regularity, in particular, through the use of optimal transport regularization. Empirical performance improvements (Section 5) are consistent with our theoretical guarantees (Section 4).

Empirically, our model consistently improves the test performance of the original models across seven diverse experiments and demonstrates the following benefits.

- **Efficiency**
    - **Impementation Efficiency:** Our optimal control framework can be directly applied to existing models, improving their performance while bypassing the costly and time-consuming trial-and-error approach to manual parameter tuning.
    - **Parameter Efficiency:** In our comprehensive experiments across various applications, we show that our framework can improve the performance of the original model with a reduced parameter count. This also improves memory efficiency during inference.

A subset of our experimental results is reported in Table 1. We show that for a variety of text generation tasks with nanoGPT and GPT-2, our OT-Transformer consistently improves the test performance using models of the same or smaller size.

Table 1: Results for text generation experiments in Section 5.4. Due to space constraints, additional LLM metrics are provided in Appendix I; our method consistently outperforms the baseline across all reported metrics.

| Experiment | Method | Para. Count | Test Loss |
|---|---|---|---|
| nanoGPT on Shakespeare (Char.-level) | Baseline | 10.65M | $2.68 \pm 0.006$ |
| | OT-Trans. (Ours) | 6.16M | $\mathbf{1.44 \pm 0.005}$ |
| GPT-2 on Shakespeare (Word-level) | Baseline | 123.7M | $5.18 \pm 0.032$ |
| | OT-Trans. (Ours) | 123.7M | $\mathbf{4.96 \pm 0.012}$ |
| GPT-2 on OpenWebText (9B tokens) | Baseline | 123.7M | 3.21 |
| | OT-Trans. (Ours) | 123.7M | $\mathbf{2.91}$ |

The empirical findings are further supported by theoretical analysis in Section 4. Our main theorem, Theorem 2, which we present a simplified version below, shows that OT-Transformer exhibits *stable forward propagation* [39, 73].

**Theorem** (Stable Forward Propagation). *For input-target output pairs* $(\mathbf{x}_1, \mathbf{y}_1)$ *and* $(\mathbf{x}_2, \mathbf{y}_2)$*, the corresponding output of the optimally trained model* $\tilde{\mathbf{y}}_1$ *and* $\tilde{\mathbf{y}}_2$ *satisfy*

$$\|\tilde{\mathbf{y}}_1 - \tilde{\mathbf{y}}_2\| \leq C\|\mathbf{x}_1 - \mathbf{x}_2\| + C'\|\mathbf{y}_1 - \mathbf{y}_2\|, \tag{1}$$

*where* $C, C' > 0$ *are constants that can be controlled by adjusting the strength of the regularization. In the absence of the regularization, these constants become unbounded, and the model can exhibit unstable forward propagation.*

This stability induces highly favorable properties in practice, highlighted as follows.

- **Robustness**

  - **Robustness to Input Perturbations:** On a data level, (1) controls the extent to which input perturbations are amplified in the output. As a result, this enhances robustness against input perturbations, such as those caused by noise or adversarial attacks.

  - **Distributional Robustness:** On a distribution level, (1) guarantees that distributional perturbations in the input space are not disproportionately amplified at the output level, which we show in Theorem 3. Distributional robustness induces robustness to training data perturbation. In particular, it ensures that a model trained on perturbed data maintains good predictive performance on the original data, with the prediction error controlled by the degree of the perturbation.

- **Generalization**

  - **In-Distribution Generalization:** Stable forward propagation improves generalization to test inputs resembling the training data, as (1) ensures that similar inputs are mapped to similar outputs. Also, distributional robustness ensures that model trained on the training distribution (sampled from the true distribution) can generalize well to data drawn from the true distribution. Combined with distributionally robust optimization (DRO), this robustness result produces non-asymptotic generalization bounds for the learned Transformer model [24, 75, 76].

  - **Out-of-Distribution Generalization:** Stable forward propagation combined with DRO also provides bounds on expected test loss for distributions within a Wasserstein ball around the empirical training distribution, offering principled insight into the model's performance on out-of-distribution samples.

We summarize our contributions as follows:

- We analyze the training and architecture of Transformers using optimal control theory. To the best of our knowledge, this is the first such analysis of Transformers from this perspective.

- Based on optimal control methods, we propose a *plug-and-play* model called OT-Transformer (8)-(9), which formulates Transformers in a continuous-time setting and incorporates an OT regularization into the training objective. Our framework is versatile, allowing for easy adaptation of existing Transformer architectures. We remark that previous attempts to apply OT to the design of Transformer models remain underexplored and have achieved limited success [5].

- Our theoretical analysis demonstrates that the OT-Transformer model confers highly beneficial theoretical properties. The most notable property is its *stable forward propagation*, which induces generalization and robustness of the learned model. These properties are further supported by our experimental results. We emphasize that our use of optimal control theory represents only an initial step in analyzing Transformers, and our framework provides a new foundation for future theoretical extensions to many other aspects of these models.

- We conduct seven extensive experiments across different fields and applications. Our framework improves the performance of the original architecture. In particular, it yields better test performance and is better at avoiding overfitting while using a reduced number of parameters.

## 2 Background

This section provides background on Transformers and continuous-time neural networks.

**Notation.** We use bold uppercase letters (e.g., $\mathbf{X}$) to denote matrices and bold lowercase letters (e.g., $\mathbf{x}$) to denote vectors. We also use $\mathbf{x}_j$ (resp. $\mathbf{x}_{i,j}$) to represent the $j$th column of $\mathbf{X}$ (resp. $\mathbf{X}_i$).

**Transformers.** Let $\{\mathbf{z}_i\}_{i=1}^n$, where $\mathbf{z}_i \in \mathbb{R}^{d_f}$, be a sequence of $n$ vectors. In the language of Transformers, each vector $\mathbf{z}_i$ is referred to as a *token*. Transformers are mappings which take in inputs $\mathbf{Z} = [\mathbf{z}_1, \ldots, \mathbf{z}_n] \in \mathbb{R}^{d_f \times n}$, with the specific form of the output depending on the downstream application. Each token is first embedded into $\mathbb{R}^d$ through the mapping

$$\mathbf{x}_{0,j} = g_l(\mathbf{z}_j; \boldsymbol{\gamma}_l), \quad \text{for} \quad j = 1, 2, ..., n, \tag{2}$$

where $\mathbf{x}_{0,j} \in \mathbb{R}^d$. The input embedding $g_l$, parametrized by weights $\boldsymbol{\gamma}_l$, embeds each token into a $d$-dimensional space and incorporates positional encoding into each token. Then, it is processed through a series of Transformer blocks, where the output of each block serves as the input to the next. At each step, the model sequentially applies the operation $\mathbf{X}_{i+1} = f_{i+1}(\mathbf{X}_i)$ where $f_{i+1} : \mathbb{R}^d \to \mathbb{R}^d$ is given by[1]

$$\mathbf{u}_{i,j} = \mathbf{x}_{i,j} + \sum_{h=1}^{H} \mathbf{W}_i^h \mathbf{V}_i^h \mathbf{X}_i \, \text{softmax}\left( \frac{(\mathbf{K}_i^h \mathbf{X}_i)^\top \mathbf{Q}_i^h \mathbf{x}_{i,j}}{\sqrt{k}} \right), \tag{3}$$

$$\mathbf{x}_{i+1,j} = \mathbf{u}_{i,j} + g_f(\mathbf{u}_{i,j}; \boldsymbol{\theta}_i), \tag{4}$$

for $j = 1, 2, ..., n$, and $i = 0, 1, ..., D - 1$, where $D$ is the total number of Transformer blocks. Here, each summand of (3) are called *self-attention heads* where $H$ is the number of heads, $\mathbf{Q}_i^h, \mathbf{K}_i^h, \mathbf{V}_i^h \in \mathbb{R}^{k \times d}$ are known as query, key, and value matrices, and $\mathbf{W}_i^h \in \mathbb{R}^{d \times k}$ are weight matrices. In (4), a fully connected layer $g_f : \mathbb{R}^d \to \mathbb{R}^d$, parametrized by weights $\boldsymbol{\theta}_i$, is applied individually to each of the $n$ tokens. This layer is common for all embedded tokens but changes for each $i$. The first equation (3) is known as multihead self-attention layers and is the key feature of Transformer architectures. A detailed derivation of (3) is given in Appendix D. This self-attention mechanism enables the model to focus on the most relevant parts of an input sequence of tokens, adapting dynamically to the context. The model can flexibly capture complex, long-distance relationships in sequential data. Such features make Transformers particularly powerful for tasks such as language processing and image recognition. In addition, the self-attention mechanism can be implemented efficiently as the manner in which the matrices are applied to process input data can be done in parallel, rendering them particularly effective for handling long sequences of tokens, i.e., when $n$ is large. In encoder-only and encoder-decoder setups, the series of Transformer blocks is referred to as the encoder; in decoder-only setups, it is referred to as the decoder.

For sequence generation tasks, the final output $\mathbf{X}_D$ is then either passed to a decoder (in the encoder-decoder setup), which consists of another series of Transformer blocks and then a multilayer perceptron (MLP). More commonly, the output feeds directly to an MLP (in the encoder-only or decoder-only setup) in downstream tasks including sequence generation, classification, or regression. Irrespective of the setup, the Transformer output $\tilde{\mathbf{y}}$ is given by

$$\tilde{\mathbf{y}} = g_o(\mathbf{X}_D; \boldsymbol{\gamma}_o), \tag{5}$$

where $g_o$ is either the composition of a decoder and an MLP or just an MLP, parametrized by $\boldsymbol{\gamma}_o$.

**Continuous-time Neural Networks.** Continuous-time models use neural networks to define the dynamics of the model's hidden states. In particular, the hidden state $\mathbf{x}(t)$ evolves according to

$$\frac{d\mathbf{x}(t)}{dt} = f_{\text{NN}}(\mathbf{x}(t), t), \tag{6}$$

where $t \in [0, T]$. The model takes $\mathbf{x}(0)$ as input and produces $\mathbf{x}(T)$ as output, with the velocity field parametrized by the neural network $f_{\text{NN}}$. Various architectures have been proposed and successfully

---

[1]Layer normalization is commonly applied in each Transformer block [92]. We omit layer normalization in this exposition for brevity, but it is included in our experiments.

applied in [15, 26, 40, 64, 84, 95, 99]. A notable and relevant advantage of continuous-time architectures is their parameter efficiency, as the continuous formulation allows them to model complex transformations over time with fewer parameters compared to discrete models. However, the use of Transformers in a continuous-time setting remains largely unexplored, with existing approaches showing limited success.

# 3   OT-Transformers

In this section, we introduce OT-Transformer, a model that can be flexibly combined with existing Transformers. The model inserts a given Transformer into a continuous-time architecture and incorporates an optimal transport (OT)-regularization into the training objective, which are motivated from the theoretical analysis using optimal control theory in Section 4.

**Model Formulation.**   Given an input sequence $\mathbf{Z} = [\mathbf{z}_1, \mathbf{z}_2, ..., \mathbf{z}_n]$ of length $n$, we first apply (2) to obtain the embedded input $\mathbf{X}_0 \in \mathbb{R}^{d \times n}$. The dynamics of the hidden state are then governed by the dynamical system

$$\frac{d\mathbf{X}(t)}{dt} = f(\mathbf{X}(t), t; \boldsymbol{\theta}), \quad \text{for} \quad t \in [0, T], \quad \text{with} \quad \mathbf{X}(0) = \mathbf{X}_0, \tag{7}$$

where $f$ is the composition of a sequence of Transformer blocks defined in (3) and (4), that is, $f = f_D \circ ... \circ f_1$, and $\boldsymbol{\theta}$ collectively denotes their trainable parameters $\boldsymbol{\theta}_i$, $\mathbf{K}_i^h$, $\mathbf{V}_i^h$, $\mathbf{Q}_i^h$ and $\mathbf{W}_i^h$ for all $h$ and $i$. Finally, we obtain the Transformer output $\tilde{\mathbf{y}}$ by applying (5) to the terminal state $\mathbf{X}(T)$.

**Plug-and-Play Formulation.**   We formulate the discretized training problem as

$$\min_{\boldsymbol{\theta}, \boldsymbol{\gamma}} \mathbb{E}_{(\mathbf{X}_0, \mathbf{y})} \left\{ G(\mathbf{X}_M, \mathbf{y}; \boldsymbol{\gamma}) + \frac{\lambda \Delta t}{2} \sum_{m=0}^{M-1} \|f(\mathbf{X}_m, t_m; \boldsymbol{\theta})\|_F^2 \right\} \tag{8}$$

$$\text{subject to} \quad \mathbf{X}_{m+1} = \mathbf{X}_m + \Delta t \cdot f(\mathbf{X}_m, t_m; \boldsymbol{\theta}), \quad m = 0, 1, \ldots, M-1. \tag{9}$$

Here, we adopt a discretize-then-optimize approach [64, 65], where (9) represents the discretized form of the continuous dynamics in (7), obtained by splitting the time interval $[0, T]$ into $M$ uniform steps with step size $\Delta t = T/M$ and $t_m = m\Delta t$. The optimization problem (8)-(9) is the discretization of the continuous-time training problem (10) over a parametrized family of Transformer models. The expectation is taken over the embedded input-output pairs $(\mathbf{X}_0, \mathbf{y})$, $\| \cdot \|_F$ denotes the Frobenius norm, $\boldsymbol{\gamma}$ collectively denotes the weights of the input embedding $\boldsymbol{\gamma}_l$ and output layer $\boldsymbol{\gamma}_o$. The loss function $G$ measures the discrepancy between the target output $\mathbf{y}$ and model output $\tilde{\mathbf{y}}(\mathbf{X}_M; \boldsymbol{\gamma})$ in (5). For instance, in classification [22] and sequence generation [83] tasks, one commonly uses the softmax loss. The second term is an OT regularization penalizing the squared norm of the velocity (the right hand side of the dynamical system (7)) at every time step. It enhances the regularity of the hidden state dynamics (7); see Section 4. We highlight that our model is *plug-and-play* and easy to implement in the sense that it can be flexibly applied to almost any existing Transformer architectures. In particular, it can directly reuse the architecture of an existing Transformer's input embedding, encoder/decoder and output layers and use its Transformer blocks $f_i$'s to construct the dynamical system (9). In practice, the OT regularization term is computed efficiently and incurs negligible overhead, as it is calculated alongside each $\mathbf{X}_m$ in (9). A schematic illustration of our model is shown in Figure 1. The regularization parameter $\lambda$ balances the effects of the two terms. Our model generalizes the vanilla formulation of Transformer blocks to continuous-time. Specifically, when $T = 1$ and $M = 1$, our model is consistent with the original discrete Transformer formulation.

On one hand, the training formulation (8) is derived from the optimal transport problem arising from the Benamou-Brenier formulation [6], as shown in Appendix E. The derivation reveals that for an appropriate choice of $\lambda > 0$, the solution to the training problem corresponds to the optimal solution of the Benamou–Brenier problem. On the other hand, we will demonstrate in the next section that the training problem can be understood through optimal control theory. And our extensive theoretical analysis shows that our model confers highly advantageous properties.

# 4 Theoretical Analysis

We provide theoretical analysis demonstrating the benefits of our model. The theoretical benefits presented in this section and the empirical evidence in Section 5 elucidate and support each other.

**Assumptions and Practical Relevance.** Our assumptions specified in Appendix F.1, are realistic in that they hold for encoder-only and decoder-only Transformer models in the continuous-time settings. They are also general and hold for other continuous-time models, including Neural ODEs [15]. These settings cover many leading models, including Vision Transformers [22], and language models including BERT [21], the GPT series [10, 69, 70] of OpenAI, PaLM [18], GLaM [23], and LaMDA [79] of Google, OPT [100] of Meta AI, and Granite of IBM [38], among others.

One of our key assumptions in deriving the theoretical advantages is for the loss function $G$ to be *convex* with respect to its first argument. Surprisingly, this holds for encoder-only and decoder-only Transformer architectures; see Appendix F.1. Generally, the convexity does not hold for encoder-decoder models. We remark that this distinction is not just theoretical, it is consistent with empirical findings from the literature. In particular, decoder-only models can match or even surpass the performance of encoder-decoder models for various large-scale language modeling tasks [28, 88, 94], despite having a simpler architecture. Moreover, current industry trends [1, 11, 80, 81] also suggest that decoder-only Transformers are replacing encoder-decoder models for LLMs. Meanwhile, in areas such as image classification or time series forecasting, encoder-only models remain a popular option, as pointed out in [41, 49, 97]. The alignment between our theoretical analysis and reported empirical performance further shows that our assumptions are relevant in practice. Our analysis offers a possible explanation for the performance gap between encoder-only/decoder-only models and encoder-decoder models, a distinction that arises from our convexity assumption. For more details, see Appendices A and F.1.

**Non-parametric Formulation.** We consider a continuous-time non-parametric formulation of (8)

$$\min_f \ \mathbb{E}_{(\mathbf{X}_0, \mathbf{y})} \left\{ G(\mathbf{X}(T), \mathbf{y}) + \frac{\lambda}{2} \int_0^T \|f(\mathbf{X}(t), t)\|_F^2 \, dt \right\}, \quad \text{subject to (7),} \tag{10}$$

where the velocity $f$ is optimized over some admissible class of functions instead of model weights. The analysis of the non-parametric formulation is justified by the universal approximation property of Transformers [98, Theorem 3], which ensures that, with enough model complexity, they can represent the optimal velocity. A central component of our approach is that the non-parametric formulation can be equivalently cast as a *mean-field control problem*, enabling theoretical analysis via optimal control theory. In the following, we highlight key theorems that show the benefits of OT-Transformer, including well-posedness of the training objective, regularity of the learned Transformer, and non-asymptotic generalization bounds. For detailed results and proofs, see Appendices F and H.

**Well-posedness of Training Problem.** We first show that the training problem is ill-posed without regularization and that the OT regularization renders the problem well-posed.

**Theorem 1.** *There exists a unique solution to the optimization problem* (10) *if and only if* $\lambda > 0$.

The ill-posedness of (10) that arises when $\lambda = 0$ is because there exists *infinitely* many velocities that are optimal. Among the infinitely many optimal velocities for the unregularized problem, some are highly irregular or have arbitrarily large magnitudes. This phenomenon has been noted in flow-based generative modeling [26, 64]. In particular, highly irregular velocities fields can produce winding hidden state trajectories which can pose challenges in numerical integration and result in numerical instability during training, as demonstrated by our experiments.

Our analysis is sufficiently general to apply to other continuous-time models beyond the Transformer architecture. In particular, Theorem 1 provides insight into the training of any neural network model under a continuous-time formulation. In the absence of regularization, such training is, in general, ill-posed. OT regularization is only one strategy that can ensure well-posedness, and optimal control methods can be used to construct other regularizations that ensures well-posedness [99].

A detailed proof is provided in Appendix F.4. The core idea is to inspect the optimality conditions for the training problem (10), which is a *Hamilton-Jacobi-Bellman* (HJB) partial differential equation

(PDE) coupled with a continuity equation. When $\lambda = 0$, the HJB PDE is not well-defined. On the other hand, $\lambda > 0$ if and only if the HJB PDE is well-defined, and the well-posedness of the resulting PDE system is, in turn, equivalent to the well-posedness of the training problem [7, 51].

**Stable Forward Propagation.** Furthermore, we show that $\lambda > 0$ not only renders the optimal velocity unique, but also guaranteed that it is highly regular; see Appendices F.5 and F.6 for details. Leveraging this regularity, we prove our main result: stable forward propagation.

**Theorem 2.** *Assume that $\lambda > TL^2$, where $T$ is the terminal time and $L$ is the 2-operator norm of the output layer weights (5). Under the assumptions in Appendix F.1, for embedded input-output pairs $(\mathbf{X}_1(0), \mathbf{y}_1)$ and $(\mathbf{X}_2(0), \mathbf{y}_2)$, the corresponding model outputs $\tilde{\mathbf{y}}_1$ and $\tilde{\mathbf{y}}_2$ satisfy*

$$\|\tilde{\mathbf{y}}_1 - \tilde{\mathbf{y}}_2\|_2 \leq \left(1 - \frac{TL^2}{\lambda}\right)^{-1} \left(L\|\mathbf{X}_1(0) - \mathbf{X}_2(0)\|_F + \frac{TL^2}{\lambda}\|\mathbf{y}_1 - \mathbf{y}_2\|_2\right). \tag{11}$$

This shows that the model, as a function of the embedded input and target output data, is Lipschitz continuous. In particular, any perturbation in the input leads to a uniformly and proportionally bounded change in the model output. This theorem also informs the selection of the hyperparameter $\lambda > 0$. On one hand, it should not be too large that the model outputs are insensitive to the input. On the other hand, it should be large enough to satisfy $\lambda > TL^2$. In practice, the condition $\lambda > TL^2$ can be enforced, and the bound (11) can be controlled through increasing $\lambda$ or imposing a weight-decay regularization on the output layer weights (5), which reduces its operator norm[2] $L$.

The proof of Theorem 2 is given in Appendix F.7. The core mathematical novelty of our paper is the use of regularity theory of HJB PDEs to prove stable forward propagation of the Transformer map in Theorem 2, which yields the distributional robustness result in Theorem 3; see Appendices F.5 and F.6. Crucially, we also use the regularity assumptions on $G$, such as its convexity.

**Generalization and Distributional Robustness.** Stable forward propagation informs out-of-distribution performance, generalization bounds, and the robustness of learned Transformers to adversarial attacks. For each target output $\mathbf{y}$, we study the regularity of the flow map of the learned dynamical system that evolves inputs $\mathbf{X}(0)$ to predictions $\tilde{\mathbf{y}}$. In this setting, Theorem 2 implies that the flow map is Lipschitz with constant $L(1 - \lambda^{-1}TL^2)^{-1}$. This result shows that while the map is not explicitly enforced to be Lipschitz during training, the OT regularization nevertheless induces this property. The Lipschitzness of the learned map implies distributional robustness, which we state formally as follows.

**Theorem 3.** *Denote the pushforward operator under the trained Transformer to be $\mathbf{T}_\sharp$ and $W_p^p(\mu, \nu) = \inf_{\gamma \in \Gamma(\mu, \nu)} \int_{\mathbb{R}^d \times \mathbb{R}^d} \|x - x'\|_p^p d\gamma$ to be the $p$-Wasserstein distance. Under the assumptions of Theorem 2, there exists a constant $\hat{C}(p) > 0$ that depends on $p$ such that for any $p \geq 1$ and any distributions $\mu$ and $\nu$,*

$$W_p(\mathbf{T}_\sharp\mu, \mathbf{T}_\sharp\nu) \leq \hat{C}(p)L\left(1 - \frac{TL^2}{\lambda}\right)^{-1} W_p(\mu, \nu). \tag{12}$$

This is a classic optimal transport result [86, Chapter 6], which holds under the Lipschitzness of the trained model proved in Theorem 2. See Appendix H.1 for a detailed proof. The bound (12) quantifies how perturbations to the input distributions $\nu$ and $\mu$ propagate to the model output distributions. The perturbations can be due to noise, new data, or adversarial modifications. Here, Wasserstein distances are crucial, as they are able to measure discrepancies between empirical distributions, which is not possible for probability divergences (e.g., Kullback-Leibler, or $f$-divergences). Crucially, (12) also depends on the *convexity* assumption on $G$ and the magnitude of OT regularization.

The bounds (11) and (12) induce highly desirable practical properties—namely, *robustness and generalization*. Due to space constraints, we refer the reader to the discussion in Section 1. Explicit formulas and detailed derivations are provided in Appendix H.

---

[2]The squared Frobenius norm of the weight matrix equals to the sum of its squared singular values, while the operator norm of the matrix is its largest singular value. Therefore, applying a weight decay regularization can decrease its operator norm $L$.

# 5  Experimental Results

We demonstrate the effectiveness of OT-Transformers through seven experiments spanning diverse applications, including point cloud classification, image classification, sentiment analysis, and text generation. Our model performs competitively and generalizes well across all tasks.

For each task, we use commonly adopted Transformer models as baselines, including encoder-only, decoder-only, and encoder-decoder architectures. This ensures a comprehensive evaluation. Moreover, while our theoretical analysis provides guarantees for encoder-only and decoder-only architectures, the empirical results demonstrate that our model also performs well with encoder-decoder architectures. We base our experiments on the setups from [48, 74], building on their code and closely following their experimental protocols. Hyperparameters, including model architectures, number of training epochs, learning rates, and layer normalization, closely follow the original setups.

For the OT-Transformer, we employ the same architectures as the baselines but with a reduced hidden dimensions, number of attention heads, and/or number of Transformer blocks. Through this, we demonstrate that OT-Transformers can achieve better performance across various tasks while having reduced model sizes. To demonstrate the effectiveness of the OT regularization, we also perform the experiments with $\lambda = 0$ in (8), effectively creating an unregularized model. We label this model unregularized OT-Transformer in the reported results.

See Appendix I for experimental details. The source code is publicly available at `https://github.com/KelvinKan/OT-Transformer`.

## 5.1  Point Cloud Classification

We use the ModelNet 40 dataset [90], which is among the most widely used benchmark for point cloud classification [82]. The dataset contains roughly 10,000 Computer-Aided Design (CAD) models that are categorized into 40 distinct classes, including common objects such as airplanes, cars, and furniture. We experiment with the Set Transformer model [53], which notably has an encoder-decoder architecture.

## 5.2  Image Classification

To further demonstrate the applicability of our proposed method, we perform experiments on imaging tasks. We use the Vision Transformer (ViT) [22], which is an encoder-only model. Since then, the model and its variants have achieved state-of-the-art performance in computer vision tasks [71, 91]. The key feature of ViTs is that they divide an image into fixed-size patches, which are treated as sequences of data. ViTs then apply self-attention mechanisms to capture relationships between these patches, enabling it to learn complex structures across the entire image. We perform two image classification experiments following the same setup as in [74].

**MNIST Classification.**  We first conduct a small-scale image classification experiment with the MNIST dataset [52]. The dataset consists of hand-written digit images, with 50,000 images used for training and 10,000 images reserved for testing. Each image is of size 28 by 28.

**Cats and Dogs Classification.**  We perform experiments on a binary cats and dogs image classification task, following [74]. The dataset contains $25,000$ training samples and $12,500$ test samples, each image in the dataset is an RGB image of size $460 \times 320$.

## 5.3  Sentiment Analysis

We perform sentiment analysis on the IMDb movie review dataset [59], which aims to predict whether each movie review is positive or negative. The dataset is balanced and contains a total of $50,000$ different reviews. The model used in the experiment is an encoder-only Transformer [74].

## 5.4  Text Generation

To further demonstrate the capabilities of the OT-Transformer, we conduct experiments on text generation. We use nanoGPT [48] and GPT-2 [69], both of which are decoder-only models with

Table 2: Results for experiments from Sections 5.1 to 5.3.

| Experiment | Method | Para. Count | Test Accuracy |
|---|---|---|---|
| Point Cloud Classification | Baseline | 0.86M | $87.4\% \pm 0.45\%$ |
| | OT-Trans. (Ours) | 0.65M | $\mathbf{89.9\% \pm 0.42\%}$ |
| Image Classification (MNIST) | Baseline | 93K | $93.0\% \pm 0.69\%$ |
| | Unreg. OT-Trans. | 18K | $96.8\% \pm 0.23\%$ |
| | OT-Trans. (Ours) | 18K | $\mathbf{97.1\% \pm 0.16\%}$ |
| Image Classification (Cats & Dogs) | Baseline | 1.77M | $77.6\% \pm 0.86\%$ |
| | Unreg. OT-Trans. | 1.48M | $78.2\% \pm 0.39\%$ |
| | OT-Trans. (Ours) | 1.48M | $\mathbf{79.0\% \pm 0.31}\%$ |
| Sentiment Analysis | Baseline | 4.74M | $83.9\% \pm 0.26\%$ |
| | Unreg. OT-Trans. | 2.37M | $82.7\% \pm 0.38\%$ |
| | OT-Trans. (Ours) | 2.37M | $\mathbf{84.6\% \pm 0.55\%}$ |

10.7 million and 124 million parameters, respectively. We conduct three different text generation experiments using different data. The goal is to evaluate the performance of OT-Transformer on text generation tasks and assess its scalability to large models with over 100 million parameters. The results are reported in Table 1 of Section 1.

**Shakespeare Dataset with nanoGPT.** We first conduct experiments using the smaller-sized nanoGPT architecture on the benchmark Shakespeare dataset. The source text is taken from Shakespeare's works, and the goal is to make predictions at the character level based on input sequences.

**Shakespeare Dataset with GPT-2.** We next perform in-depth experimentation on the Shakespeare dataset using the much larger GPT-2 architecture, which contains over 100 million trainable parameters. Note that for this experiment, token prediction is performed at the word level, making the task more difficult compared to the previous example.

**OpenWebText Dataset with GPT-2.** Lastly, to demonstrate the applicability of our model to large-scale problems, we conduct experiments using the GPT-2 architecture as a baseline on the OpenWebText dataset. This dataset, originally curated in [36] from Reddit posts, includes a training set of approximately 9 billion tokens and a validation set of around 4 million tokens. This experiment is large-scale in both model and dataset size, representing a realistic application setting.

## 5.5 Summary of Numerical Results

**Compiled Results.** We present the compiled numerical results in Tables 1 and 2, and summarize the main findings as follows. First, our proposed model demonstrates competitive performance across all seven test examples, ranging from small-scale to large-scale experiments, highlighting its ability to consistently *improve performance* over baseline models. Second, OT-Transformer *outperforms* baseline models across various examples while using significantly smaller models, showcasing its parameter efficiency. Finally, OT-Transformer avoids overfitting more effectively, resulting in improved generalization and lower test loss compared to the baseline; see Figures 5, 6 and 8 in Appendix I.

Note that while our experiments focus on generalization metrics like test loss and accuracy, strong performance in these settings also reflects robustness. In real-world scenarios, test data often differ from training data due to noise, sampling variability, or distribution mismatch. The consistent results across different experiments support our theory Theorems 2 and 3 on robustness to input, distributional and training data perturbations.

**Robustness Tests.** To further validate our stability theory, we conduct four additional experiments evaluating the performance of OT-Transformer on test data corrupted by varying levels of random noise. The noise was absent from the training data. This setup assesses the model's forward stability, distributional robustness and out-of-distribution generalization.

Table 3: Robustness tests under varying levels of noise.

| Experiment | Metric/replace rate | 0.0 | 0.005 | 0.01 | 0.05 | 0.1 |
|---|---|---|---|---|---|---|
| **NanoGPT** (text replace) | Loss (±std) Base. | $2.68_{\pm0.004}$ | $2.78_{\pm0.004}$ | $2.88_{\pm0.004}$ | $3.65_{\pm0.004}$ | $4.60_{\pm0.005}$ |
| | Loss (±std) Ours | $\mathbf{1.44_{\pm0.005}}$ | $\mathbf{1.49_{\pm0.004}}$ | $\mathbf{1.55_{\pm0.003}}$ | $\mathbf{1.95_{\pm0.010}}$ | $\mathbf{2.42_{\pm0.022}}$ |
| | Drop (↓) Base. | – | 0.10 | 0.20 | 0.97 | 1.92 |
| | Drop (↓) Ours | – | **0.05** | **0.11** | **0.51** | **0.98** |

| Experiment | Metric/drop rate | 0.0 | 0.01 | 0.05 | 0.1 | 0.2 | 0.5 |
|---|---|---|---|---|---|---|---|
| **Point cloud** (dropout) | Acc. (±std) Base. | $86.6\%_{\pm0.45\%}$ | $86.6\%_{\pm0.48\%}$ | $85.8\%_{\pm0.60\%}$ | $84.3\%_{\pm0.69\%}$ | $76.9\%_{\pm0.88\%}$ | $34.5\%_{\pm1.94\%}$ |
| | Acc. (±std) Ours | $\mathbf{89.3\%_{\pm0.55\%}}$ | $\mathbf{89.3\%_{\pm0.55\%}}$ | $\mathbf{88.8\%_{\pm0.34\%}}$ | $\mathbf{87.6\%_{\pm0.62\%}}$ | $\mathbf{83.9\%_{\pm0.80\%}}$ | $\mathbf{55.4\%_{\pm4.87\%}}$ |
| | Drop (↓) Base. | – | 0.0% | 0.8% | 2.3% | 9.7% | 52.1% |
| | Drop (↓) Ours | – | **0.0%** | **0.5%** | **1.7%** | **5.4%** | **33.9%** |

| Experiment | Metric/noise level | 0.0 | 0.01 | 0.05 | 0.1 | 0.2 | 0.5 |
|---|---|---|---|---|---|---|---|
| **MNIST** (Gauss. noise) | Acc. (±std) Base. | $92.97\%_{\pm0.67\%}$ | $92.99\%_{\pm0.66\%}$ | $92.96\%_{\pm0.69\%}$ | $92.76\%_{\pm0.72\%}$ | $91.70\%_{\pm0.68\%}$ | $80.64\%_{\pm1.58\%}$ |
| | Acc. (±std) Ours | $\mathbf{97.05\%_{\pm0.15\%}}$ | $\mathbf{97.05\%_{\pm0.16\%}}$ | $\mathbf{96.99\%_{\pm0.18\%}}$ | $\mathbf{96.89\%_{\pm0.15\%}}$ | $\mathbf{96.39\%_{\pm0.11\%}}$ | $\mathbf{90.10\%_{\pm1.10\%}}$ |
| | Drop (↓) Base. | – | -0.02% | 0.01% | 0.21% | 1.27% | 12.33% |
| | Drop (↓) Ours | – | 0.00% | 0.06% | **0.16%** | **0.66%** | **6.95%** |
| **MNIST** (Uni. noise) | Acc. (±std) Base. | $92.97\%_{\pm0.67\%}$ | $92.99\%_{\pm0.65\%}$ | $92.98\%_{\pm0.63\%}$ | $92.90\%_{\pm0.58\%}$ | $92.64\%_{\pm0.57\%}$ | $90.02\%_{\pm0.45\%}$ |
| | Acc. (±std) Ours | $\mathbf{97.05\%_{\pm0.15\%}}$ | $\mathbf{97.03\%_{\pm0.14\%}}$ | $\mathbf{97.00\%_{\pm0.15\%}}$ | $\mathbf{96.97\%_{\pm0.14\%}}$ | $\mathbf{96.79\%_{\pm0.16\%}}$ | $\mathbf{95.57\%_{\pm0.11\%}}$ |
| | Drop (↓) Base. | – | 0.02% | -0.02% | 0.07% | 0.33% | 2.95% |
| | Drop (↓) Ours | – | **0.02%** | 0.05% | **0.08%** | **0.26%** | **1.48%** |

In the first experiment, we evaluate the NanoGPT model, where each character in the test data is randomly replaced at a specified rate, following the setup in [42]. In the second experiment, we perform point cloud classification under point dropout, where a fraction of points in each test sample is removed randomly, following [68, 89]. The third and fourth experiments focus on MNIST dataset with Gaussian and uniform noise added to the test images, respectively. The setups follow [37].

The results are reported in Table 3. Our model consistently outperforms the baselines over all these tests across all noise levels. Importantly, the performance gap widens at higher noise levels. These additional results again corroborate our theory and other empirical findings. They also further highlights the attractiveness of our model in practice.

We also provide an empirical sensitivity study on the model's hyperparameters. The results show that our model again consistently outperforms the baseline across various numbers of integration steps and over two orders of magnitude of the regularization strength $\lambda$. Due to space constraints; detailed results are provided in Appendix C.

Overall, the OT-Transformer results provide evidence — backed up by our theory in Section 4 — that optimal transport, acts as a unifying regularization principle across text, image, and 3D modalities. For more details, we direct readers to Appendix I.

## 6 Discussion and Summary

In this work, we analyze the Transformer architecture and training through a proposed *optimal control framework*. Based on this, we propose OT-Transformer, a plug-and-play model which can be flexibly applied to established Transformers with minimal code changes. OT-Transformer improves the performance of existing models while conferring strong theoretical guarantees. These include generalization and robustness, established through our optimal control-based analysis. We highlight that key theoretical results rely on the novel application of the regularity theory of HJB PDEs to prove stable forward propagation and distributional robustness of the learned Transformer. This further proves non-asymptotic generalization bounds through DRO. These theoretical results are supported by extensive experiments that demonstrate the effectiveness of the optimal control framework and further show that the resulting OT-Transformer model improves parameter efficiency. Overall, our framework provides a foundation for systematic and theory-driven improvements for Transformers.

We emphasize that this work represents only a first step toward building a control-theoretic foundation for designing and analyzing Transformer architectures and training. Many additional components of Transformers — such as layer normalization, attention mechanism, and others — present promising directions for future investigation through optimal control methods. In addition, exploring alternative numerical integration schemes may offer a path toward improving training efficiency; we leave this for future work.

## Acknowledgements

The authors would like to thank Lars Ruthotto (Emory University) and Tingwei Meng (Amazon Robotics) for their valuable advice and insightful discussion.

The authors would also like to thank the four anonymous reviewers for their thorough review and constructive suggestions.

K. Kan and S. Osher were partially funded by the U.S. Department of Energy (DOE), Office of Science (SC), Advanced Scientific Computing Research program under award B&R# KJ0401010, FWP# CC147.

X. Li was partially funded by NSF 2339678 and 2321040.

B. Zhang and M. Katsoulakis were partially funded by AFOSR grant FA9550-21-1-0354.

This material is based upon work of M. Katsoulakis and T. Sahai supported by the Defense Advanced Research Projects Agency (DARPA) under Agreement No. HR00112590112. Approved for public release; distribution is unlimited.

S. Osher was partially funded by DARPA under grant HR0011259007, NSF under grants 1554564 and 220827, AFOSR under MURI grant N00014-20-1-2787, and ARO under grant W911NF-24-1-0157.

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

# A    Related Work

This section provides a review of relevant work.

**Continuous-time Architecture.**    There has been some applications of continuous-time formulations of Transformers. However, we note that they do not provide theoretical analysis to justify their modeling choice or to demonstrate its advantages. In contrast, our work proposes a foundational framework for understanding and designing effective Transformer architectures grounded in optimal control theory. Moreover, our work offers extensive theoretical analysis—supported by strong empirical results across diverse experiments—that motivates and substantiates the advantages of our model. Moreover, there is a key distinction between our and existing model formulations. In OT-Transformer, we use the composition of all Transformer blocks to parametrize a single dynamical system (7) governing the hidden states. To the best of our knowledge, the existing works use each Transformer block to parametrize a dynamical system. For a Transformer with $D$ Transformer blocks, the continuous-time model is represented as the output of $D$ different dynamical systems. In particular, it is formulated as

$$
\begin{aligned}
\mathbf{X}_0(0) &= \mathbf{X}_0, \\
\mathbf{X}_i(0) &= \mathbf{X}_{i-1}(T), && \text{for } 1 \leq i \leq D-1, \\
\frac{d\mathbf{X}_i(t)}{dt} &= \hat{f}_i(\mathbf{X}_i(t), t; \hat{\boldsymbol{\theta}}_i), && \text{for } t \in [0, T], \ 0 \leq i \leq D-1,
\end{aligned}
\tag{13}
$$

where $\hat{f}_i$ is the $i$th Transformer block parametrized by weights $\hat{\boldsymbol{\theta}}_i$ and defined in (3) and (4), except that the fully-connected layer (4) has no skip-connection.

This formulation is introduced in [5]. Here, we highlight several key differences between their work and ours. First, they only conduct the simple task of determining the parity of a binary sequence in their work, rather than investigating its performance in general applications. More importantly, when their approach is applied, it fails to improve performance over the vanilla Transformer and instead degrades it. In their experiments, the optimal transport regularization cannot improve the performance of their model when the sequence length exceeds eight. We observe similar issues when testing their model on other applications; see Appendix I. This is potentially due to their choice of formulation. Specifically, in (13), as the model transitions from one Transformer block to the next, it effectively switches to a different dynamical system, introducing non-smoothness to the overall dynamics. This undermines the purpose of the optimal transport regularization and violates the regularity properties proved in Appendix F. In contrast, our model is formulated using only one dynamical system. The resulting dynamics is smoother and thus inherently better suited to incorporate the optimal transport regularization. In particular, our modeling choice is consistent with the regularity properties proved in Appendices F.5 and F.6. This is also evident in our experimental results, while the regularization can always improve the generalization of OT-Transformer to a significant extent, this is not the case with their model; in certain scenarios, the regularization may even degrade their model's performance. We also mention that, while [5] proposes alternative formulations for further investigation, it does not consider ours, highlighting the novelty and non-triviality of our approach.

Since then, there have been a number of follow-up works that build on the formulation Equation (13) to perform different tasks, including sequence generation [54, 55, 58, 101], time series forecasting [16, 93], and image classification [62, 63]. Most of these methods only use the formulation (13) as motivation, and none of them consider optimal transport regularization in their approach. Moreover, these models focused on a specific type of application and not general-purpose.

In order to access the performance of our OT-Transformer more comprehensively, we also include the existing Transformer formulation (13) as a benchmark in our experiments; see Appendix I for detailed experimental results. It is referred to as "N-ODE Transformer" in our experimental results, following the terminology in [5].

**OT-based CNFs.**    A prominent application of continuous-time neural networks is continuous normalizing flows (CNFs) [15]. CNFs use (6) to paramtrize invertible mappings between a standard Gaussian distribution and an unknown target distribution. The ill-posed nature of the CNF formulation can often add to the complexity and computational cost for solving a problem. Optimal transport (OT) based regularization has prominent applications in CNFs and is a powerful tool in improving accuracy and at times reducing cost. Among the infinitely many mappings between the two distributions,

OT-based CNFs [26, 64, 84, 95] target to find the optimal transport mapping. This is done by incorporating into the training objective regularization term(s) enforcing straight trajectories in (6). This renders the training problem well-posed [40, 99]. The straight trajectories also offer numerical advantages, as they make the numerical integration of (6) more tractable.

**Mathematical Analysis.** There have been works that theoretically analyze a continuous-time formulation of Transformers. In [33, 34], they show that a continuous-time formulation can be interpreted as an interacting particle system, where each token can be perceived as a particle. They demonstrate that there is a clustering behavior among the tokens. Since then, there has been a number of works that further investigate the dynamics of tokens through this interpretation, including [2, 8, 12, 35, 47], to name a few. However, we note that the aforementioned work is primarily theoretical and lacks evaluations beyond toy experiments. In [74], they show that, under some restriction on the weights, a continuous-time formulation of self-attention layers can be interpreted as a gradient flow. In [45], they use a continuous-time training formulation to analyze the stability of Transformers under layer normalization. However, no experiments have been conducted following this analysis. To the best of our knowledge, existing theoretical analyses have not been conducted on continuous-time Transformers with OT regularization.

**Relevance of Assumptions.** We highlight the relevance of our assumptions and their alignment with recent developments in the literature and industry. In a recent survey [94] on LLM models, since the inception of GPT-style models, there has been a clear trend of decoder-only architectures taking popularity over encoder-only and encoder-decoder architectures due to their superior performance despite having a simpler architecture, with many of the industry standard such as GPT models [1, 11], Llama [80, 81] falling under the category. Similar observation is also discussed in [28, 88]. Going beyond LLMs, survey work such as [41] suggest encoder-only models are most popular for image classification tasks. While for time series problems, encoder-only architecture has been the norm with exploration of decoder-only architectures emerging, as pointed out in [49, 97]. While our proposed approach is applicable to any architecture, these trends underscore the importance of our theoretical guarantees on both encoder-only and decoder-only models, as they continue to define the state of the art across diverse domains and tasks.

## B Comparison Between a Standard Transformer and OT-Transformer

**Implementation.** To compare between our OT-Transformer and a standard Transformer, we provide a side-by-side pseudocode of the forward propagation. This highlights the shared components, the OT-specific regularization and updates, and how OT-Transformer can directly use an existing Transformer in a plug-and-play manner.

---

**Algorithm 1** Forward Propagation: Standard Transformer vs OT-Transformer

| **Standard Transformer:** | **OT-Transformer:** |
|---|---|
| **Require:** Input $\mathbf{X}_0$, Transformer model $f = f_D \circ \ldots \circ f_1$ | **Require:** Input $\mathbf{X}_0$, Transformer model $f = f_D \circ \ldots \circ f_1$, terminal time $T$, regularization parameter $\lambda$, number of numerical integration steps $M$ |
| | **Initialize:** $\mathbf{X}(0) = \mathbf{X}_0$, reg $= 0$, $\Delta t$ |
| 1: **for** $i = 1, 2, \ldots, D$ **do** | 1: **for** $t = \Delta t, 2\Delta t, \ldots, T$ **do** |
| | 2:    Compute $f(\mathbf{X}(t))$ |
| |     % Adapt from standard Transformer |
| 2:    $\mathbf{X}_i \leftarrow f_i(\mathbf{X}_{i-1})$ | 3:    $\mathbf{X}(t + \Delta t) \leftarrow \mathbf{X}(t) + \Delta t \cdot f(\mathbf{X}(t))$ |
| | 4:    reg $\leftarrow$ reg $+ \Delta t \cdot \|f(\mathbf{X}(t))\|_F^2$ |
| 3: **end for** | 5: **end for** |
| 4: **return:** output $\mathbf{X}_D$ | 6: **return** output $\mathbf{X}(t)$, regularization $\lambda \cdot$ reg |

---

**Computational Overhead.** Since OT-Transformer performs numerical integration (corresponding to the for loop in line 1 of Algorithm 1), it incurs computational overhead compared to a standard Transformer. However, the computational cost can be mitigated by using a smaller model thanks to OT-Transformer's parameter efficiency; see Tables 1 and 2.

Moreover, the OT regularization promotes highly regular hidden states (see Appendices F.2-F.7). This allows OT-Transformer to further reduce computational cost: by using fewer integration steps (bigger $\Delta t$) without sacrificing integration accuracy; see Table 5. Consequently, OT-Transformer incurs less computational overhead than regular continuous-time Transformers. A concrete runtime comparison for the NanoGPT experiment is reported in Table 4.

Table 4: Average training time per iteration for the NanoGPT experiment.

| Model | Time (ms) |
|---|---|
| Baseline | 64.6 |
| OT-Transformer | 175.0 |
| Standard Continuous-time Transformer | 344.4 |

## C Empirical Sensitivity Study of Hyperparameters

We perform additional NanoGPT experiments to test the sensitivity of OT-Transformer with respect to its hyperparameters. The experiments are done over three random trials. The results are reported in Table 5. We see from the results that OT-Transformer consistently outperforms the baseline across over two orders of magnitude of $\lambda$. The performance of OT-Transformer is also very stable with respect to the number of time steps. Moreover, tuning of hyperparameters is generally not required; even if the optimal $\lambda$ and number of integration steps ($T/\Delta t$) are not used, OT-Transformer still outperforms the baseline.

Table 5: Ablation study on the strength of OT regularization ($\lambda$) and integration time step ($T/\Delta t$) for the NanoGPT experiment. The experiments are done over three random trials.

| $\lambda$ | 0.01 | 0.05 | 0.1 | 0.5 | 1 | 5 | **Baseline** |
|---|---|---|---|---|---|---|---|
| Test loss | $2.68 \pm 0.019$ | $2.29 \pm 0.017$ | $2.05 \pm 0.009$ | $1.52 \pm 0.002$ | $\mathbf{1.44 \pm 0.004}$ | $1.48 \pm 0.003$ | $2.68 \pm 0.006$ |
| **Time step** | 1 | 5 | 10 | 15 | 20 | **Baseline** | |
| Test loss | $1.46 \pm 0.003$ | $\mathbf{1.43 \pm 0.002}$ | $1.44 \pm 0.004$ | $1.46 \pm 0.002$ | $1.49 \pm 0.008$ | $2.68 \pm 0.006$ | |

## D Derivation of the Transformer Equation

We present the derivation of (3), the equation for the multihead self-attention layer. Recall that $\mathbf{X}_i \in \mathbb{R}^{d \times n}$ denotes the input to the $(i + 1)$th Transformer block for $i = 0, ..., D - 1$, where $n$ is the number of tokens, $d$ is their dimension, and $D$ is the total number of Transformer blocks. And $\mathbf{x}_{i,j} \in \mathbb{R}^d$ denotes the $j$th column of $\mathbf{X}_i$.

Using the notation defined in Section 2, the first four equations in [78] are given by

$$Q_i^{(h)}(\mathbf{x}_{i,j}) = \mathbf{Q}_i^h \mathbf{x}_{i,j}, \ K_i^{(h)}(\mathbf{x}_{i,j}) = \mathbf{K}_i^h \mathbf{x}_{i,j}, \ V_i^{(h)}(\mathbf{x}_{i,j}) = \mathbf{V}_i^h \mathbf{x}_{i,j}, \quad \text{where} \quad \mathbf{Q}_i^h, \mathbf{K}_i^h, \mathbf{V}_i^h \in \mathbb{R}^{k \times d},$$

(14)

$$\alpha_{j,j'}^{(h)} = \text{softmax}_{j'} \left( \frac{\langle Q_i^{(h)}(\mathbf{x}_{i,j}), K_i^{(h)}(\mathbf{x}_{i,j'}) \rangle}{\sqrt{k}} \right),$$

(15)

$$\mathbf{u}_{i,j}' = \sum_{h=1}^{H} \mathbf{W}_i^h \sum_{j'=1}^{n} \alpha_{j,j'}^{(h)} V_i^{(h)}(\mathbf{x}_{i,j'}), \qquad \text{where} \quad \mathbf{W}_i^h \in \mathbb{R}^{d \times k},$$

(16)

$$\mathbf{u}_{i,j} = \text{LayerNorm}(\mathbf{x}_{i,j} + \mathbf{u}_{i,j}').$$

(17)

Here, $H$ is the number of attention heads, and $\text{softmax}_{j'}$ denotes the softmax function applied on a $d$-dimensional vector indexed by $j'$.

Substituting (14) into (15) and (16), and plugging (16) into (17), we obtain

$$\alpha_{j,j'}^{(h)} = \text{softmax}_{j'}\left(\frac{\langle \mathbf{Q}_i^h \mathbf{x}_{i,j}, \mathbf{K}_i^h \mathbf{x}_{i,j'} \rangle}{\sqrt{k}}\right) = \text{softmax}_{j'}\left(\frac{\left(\mathbf{Q}_i^h \mathbf{x}_{i,j}\right)^\top \left(\mathbf{K}_i^h \mathbf{x}_{i,j'}\right)}{\sqrt{k}}\right)$$
$$= \text{softmax}_{j'}\left(\frac{\left(\mathbf{K}_i^h \mathbf{x}_{i,j'}\right)^\top \left(\mathbf{Q}_i^h \mathbf{x}_{i,j}\right)}{\sqrt{k}}\right), \tag{18}$$

$$\mathbf{u}_{i,j} = \text{LayerNorm}\left(\mathbf{x}_{i,j} + \sum_{h=1}^H \mathbf{W}_i^h \sum_{j'=1}^n \alpha_{j,j'}^{(h)} \mathbf{V}_i^h \mathbf{x}_{i,j'}\right)$$
$$= \text{LayerNorm}\left(\mathbf{x}_{i,j} + \sum_{h=1}^H \mathbf{W}_i^h \sum_{j'=1}^n \mathbf{V}_i^h \mathbf{x}_{i,j'} \alpha_{j,j'}^{(h)}\right). \tag{19}$$

Here, in the last step, we used the fact that $\alpha_{j,j'}^{(h)}$'s are scalars. We then plug (18) into (19) and obtain

$$\mathbf{u}_{i,j} = \text{LayerNorm}\left(\mathbf{x}_{i,j} + \sum_{h=1}^H \mathbf{W}_i^h \sum_{j'=1}^n \mathbf{V}_i^h \mathbf{x}_{i,j'} \, \text{softmax}_{j'}\left(\frac{\left(\mathbf{K}_i^h \mathbf{x}_{i,j'}\right)^\top \left(\mathbf{Q}_i^h \mathbf{x}_{i,j}\right)}{\sqrt{k}}\right)\right)$$
$$= \text{LayerNorm}\left(\mathbf{x}_{i,j} + \sum_{h=1}^H \mathbf{W}_i^h \mathbf{V}_i^h \mathbf{X}_i \, \text{softmax}\left(\frac{\left(\mathbf{K}_i^h \mathbf{X}_i\right)^\top \left(\mathbf{Q}_i^h \mathbf{x}_{i,j}\right)}{\sqrt{k}}\right)\right), \tag{20}$$

where in the last step we used the fact that $\mathbf{x}_{i,j'}$ is the $j'$th column of $\mathbf{X}_i$. Recall that in the main text, we omitted layer normalization for simplicity of exposition. The formulation (20) becomes the self-attention layer formulation (3) when we omit the layer normalization function. Thus, we have derived the multihead self-attention layer formulation (3).

## E  Optimal Transport Background and Derivation of Training Problem

We review the relevant optimal transport background and then derive the training objective (8).

**Optimal Transport Background.** We consider the space $\mathbb{R}^{n \times d}$, to which the hidden states of OT-Transformer belong. Let $\mathcal{P}_2(\mathbb{R}^{n \times d})$ be the space of Borel probability measure on $\mathbb{R}^{n \times d}$ with finite second-order moments and $\rho_0, \rho_1 \in \mathcal{P}_2(\mathbb{R}^{n \times d})$, which specify the initial and terminal distributions. The Monge-Kantorovich problem with quadratic cost [85, Chapter 1] is given by

$$W_2^2(\rho_0, \rho_1) = \inf_{\pi \in \Gamma(\rho_0, \rho_1)} \iint_{\mathbb{R}^{n \times d} \times \mathbb{R}^{n \times d}} \|\mathbf{X} - \mathbf{Y}\|_F^2 \, d\pi(\mathbf{X}, \mathbf{Y}), \tag{21}$$

where $\Gamma(\rho_0, \rho_1)$ is the set of joint probability measures with on $\mathbb{R}^{n \times d} \times \mathbb{R}^{n \times d}$ with $\mathbf{X}$- and $\mathbf{Y}$-marginal distributions $\rho_0$ and $\rho_1$, respectively. Here, the quadratic $\|\mathbf{X} - \mathbf{Y}\|_F^2$ quantifies the cost of transporting one unit of mass from location $\mathbf{X}$ to location $\mathbf{Y}$. We note that the Wasserstein space $(\mathcal{P}_2(\mathbb{R}^{n \times d}), W_2)$ equipped with the Wasserstein metric $W_2$ is a complete separable metric space [86, Theorem 6.18].

The Benamou-Brenier formulation of (21) is given by [6]

$$\inf_{f,\rho} \int_0^T \int_{\mathbb{R}^{d \times n}} \frac{1}{2} \|f(\mathbf{X}, t)\|_F^2 \, \rho(\mathbf{X}, t) \, d\mathbf{X} dt \tag{22}$$
$$\text{subject to} \quad \partial_t \rho(\mathbf{X}, t) + \nabla \cdot (\rho(\mathbf{X}, t) f(\mathbf{X}, t)) = 0, \tag{23}$$
$$\text{and} \quad \rho(\mathbf{X}, 0) = \rho_0(\mathbf{X}), \ \rho(\mathbf{X}, T) = \rho_1(\mathbf{X}). \tag{24}$$

This is a dynamic formulation of (21), which describes optimal transport as a dynamical system. In particular, the probability density $\rho : \mathbb{R}^{d \times n} \times [0, T] \to \mathbb{R}_{\geq 0}$ evolves continuously over time under the velocity field $f : \mathbb{R}^{d \times n} \times [0, T] \to \mathbb{R}^{d \times n}$, as governed by the continuity equation (23). The initial and terminal conditions (24) require that the probability density evolve from $\rho(\mathbf{X}, 0) = \rho_0(\mathbf{X})$

to $\rho(\mathbf{X}, T) = \rho_1(\mathbf{X})$. The optimal velocity field has several important and favorable properties. Mass induced by the optimal velocity travels in straight lines at constant speed [86][Corollary 7.22]. Moreover, under standard conditions on $\rho_0$ and $\rho_1$, the optimal velocity field is unique [9, 50].

Since the velocity field $f$ governs the movement of mass, given an initial position $\mathbf{X}_0 \sim \rho_0$, the evolution of $\mathbf{X}(t)$ is governed by the ODE

$$\frac{d\mathbf{X}(t)}{dt} = f(\mathbf{X}(t), t), \quad \text{for} \quad t \in [0, T], \quad \text{with} \quad \mathbf{X}(0) = \mathbf{X}_0. \tag{25}$$

Denote the solution operator of (25) by $\mathcal{S} : \mathbb{R}^{n \times d} \times [0, T] \to \mathbb{R}^{n \times d}$ such that $\mathcal{S}(\mathbf{X}_0, t) = \mathbf{X}(t)$. Under suitable regularity conditions [3][Lemma 8.1.6], the solution of the continuity equation (23) is given by $\rho(\cdot, t) = \mathcal{S}(\cdot, t)_{\#}\rho_0$, the pushforward of the probability measure $\rho_0$ by $\mathcal{S}(\cdot, t)$. Hence, (22)-(24) can be rewritten to[3]

$$\inf_f \int_0^T \int_{\mathbb{R}^{d \times n}} \frac{1}{2}\|f(\mathbf{X}(t), t)\|_F^2 \, \rho_0(\mathbf{X}_0) \, d\mathbf{X}_0 dt$$

$$\text{subject to} \quad \frac{d\mathbf{X}(t)}{dt} = f(\mathbf{X}(t), t) \quad \text{for} \quad t \in [0, T], \quad \mathbf{X}(0) = \mathbf{X}_0, \quad \mathcal{S}(\cdot, T)_{\#}\rho_0 = \rho_1. \tag{26}$$

**Derivation of Training Problem.** Under the setup of OT-Transformer, $f$ is the Transformer blocks, and $\mathbf{X}_0$ and $\rho_0$ are the embedded input and its distribution, respectively. The probability measure $\rho_1$ specifies the target distribution of the terminal state $\mathbf{X}(T)$. The terminal condition of (26) requires that the distributions of the target output $\mathbf{y}$ and corresponding terminal state $\mathbf{X}(T)$ to match. That is, it requires $G(\mathbf{X}(T), \mathbf{y}) = 0$[4] for each pair $(\mathbf{X}(T), \mathbf{y})$, where $G$ is the loss function defined in (8). Thus, (26) becomes

$$\inf_f \mathbb{E}_{\mathbf{X}_0, \mathbf{y}} \int_0^T \frac{1}{2}\|f(\mathbf{X}(t), t)\|_F^2 \, dt$$

$$\text{subject to} \quad \frac{d\mathbf{X}(t)}{dt} = f(\mathbf{X}(t), t) \quad \text{for} \quad t \in [0, T], \quad \mathbf{X}(0) = \mathbf{X}_0, \quad G(\mathbf{X}(T), \mathbf{y}) = 0. \tag{27}$$

Here, we used Fubini's theorem [77, Theorem 3.1] to swap the order of the integrations, the expectation is taken over the joint distribution of the input-output pairs $(\mathbf{X}_0, \mathbf{y})$ with an $\mathbf{X}_0$-marginal distribution $\rho_0$. Further, we make the following remarks on (27):

- The optimization is over $f$ which is non-parametric. In OT-Transformer, we parametrize $f$ using Transformer blocks with weights $\boldsymbol{\theta}$, and the weights of the embedding and output layers are $\boldsymbol{\gamma}$.

- The optimization problem is intractable in general when the terminal condition $G(\mathbf{X}(T), \mathbf{y}) = 0$ is imposed as a hard constraint. We relax this constraint by incorporating it as a penalty term in the objective function.

Thus, we obtain the training problem

$$\min_{\boldsymbol{\theta}, \boldsymbol{\gamma}} \mathbb{E}_{\mathbf{X}_0, \mathbf{y}} \left\{ \mu G(\mathbf{X}(T), \mathbf{y}; \boldsymbol{\gamma}) + \int_0^T \frac{1}{2}\|f(\mathbf{X}(t), t; \boldsymbol{\theta})\|_F^2 \, dt \right\}$$

$$\text{subject to} \quad \frac{d\mathbf{X}(t)}{dt} = f(\mathbf{X}(t), t; \boldsymbol{\theta}) \quad \text{for} \quad t \in [0, T], \quad \mathbf{X}(0) = \mathbf{X}_0. \tag{28}$$

Here, when we set $\mu = \frac{1}{\lambda}$, under discretization (28) is equivalent to the OT-Transformer training problem (8) in the continuous-time setting subject to the dynamics (7). It is noteworthy that the hyperparameter $\mu$ can be interpreted as the Lagrange multiplier for the terminal condition of (27). This reveals that the regularization hyperparameter $\lambda$ is inversely proportional to the Lagrange multiplier. When we have $\lambda = \frac{1}{\mu^*}$, where $\mu^*$ is the optimal Lagrange multiplier, and assuming that the Transformer blocks parametrizing $f$ are sufficiently expressive, the solution of the OT-Transformer training problem (8) corresponds to the solution of the hard-constrained problem (27).

---

[3]For clarity of presentation, we slightly abuse notation by using $\mathbf{X}_0$ both as a random variable and as a dummy variable of integration.

[4]For the commonly used cross-entropy [46][Section 1] and mean-squared error, the loss is zero when the model output equals the target output $\mathbf{y}$.

# F Proofs of Theorems

In this section, we report in detail the assumptions and derivations of the theorems in Section 4.

## F.1 Assumptions and Justification

In the following, we list the assumptions of the training problem (8), on which our theoretical analysis is based. We then justify our assumptions by showing that they are satisfied in common applications.

1. the function $G$ is proper, convex, and twice continuously differentiable in its first argument;
2. the function $\nabla G(\cdot, \cdot)$ is Lipschitz continuous in both arguments, where the gradient is taken with respect to its first argument.

Here, the function $G$ is the first term in the training objective (8), which is the composition of the output layer (5) and the loss function.

The assumptions are valid for encoder-only and decoder-only continuous-time Transformer training problems for classification, next-token prediction, and regression tasks. These settings apply to many common Transformer architectures in the continuous-time settings, including Vision Transformers [22], and language models including the GPT series [10, 69, 70] of OpenAI, PaLM [18], GLaM [23], and LaMDA [79] of Google, OPT [100] of Meta AI, and Granite of IBM [38].

Next, we will demonstrate in detail why the assumptions hold under these settings. In this subsection, for clarity of presentation, we vectorize the terminal hidden states and denote their vectorizations by $\mathbf{x}(T) = \text{vec}(\mathbf{X}(T))$. Accordingly, we denote functions originally defined on matrices (e.g., $G(\mathbf{X}(T), \mathbf{y})$) by functions of their vectorized forms (e.g., $G(\mathbf{x}(T), \mathbf{y})$). This is a slight abuse of notation, which will significantly simplify expressions and derivations.

On one hand, for regression tasks, the output layer (5) consists of a linear transformation

$$\hat{\mathbf{y}} = \boldsymbol{\psi}_o \mathbf{x}(T), \tag{29}$$

where $\boldsymbol{\psi}_o \in \mathbb{R}^{c \times dn}$ are the weights of the output layer. The loss function is the mean-squared loss (MSE)

$$\text{MSE}(\hat{\mathbf{y}}, \mathbf{y}) = \frac{1}{2}\|\hat{\mathbf{y}} - \mathbf{y}\|_2^2. \tag{30}$$

Given a target output $\mathbf{y} \in \mathbb{R}^c$, the loss function $G$ is given by

$$\begin{aligned} G(\mathbf{x}(T), \mathbf{y}) &= \text{MSE}(\boldsymbol{\psi}_o \mathbf{x}(T), \mathbf{y}) \\ &= \frac{1}{2}\|\boldsymbol{\psi}_o \mathbf{x}(T) - \mathbf{y}\|_2^2. \end{aligned} \tag{31}$$

It is easy to see that $G$ is proper, convex, and smooth (hence twice continuously differentiable). This satisfies the first assumption. Its gradient is given by

$$\nabla G(\mathbf{x}(T), \mathbf{y}) = \boldsymbol{\psi}_o^\top (\boldsymbol{\psi}_o \mathbf{x}(T) - \mathbf{y}), \tag{32}$$

which is Lipschitz continuous in each of its two arguments, with Lipschitz constants $L^2$ and $L$, respectively, where

$$L = \|\boldsymbol{\psi}_o\|_2 = \|\boldsymbol{\psi}_o^\top\|_2, \tag{33}$$

with $\|\boldsymbol{\psi}_o\|_2$ denoting the 2-operator norm of $\boldsymbol{\psi}_o$. And we used the fact that $\|\boldsymbol{\psi}_o\|_2 = \|\boldsymbol{\psi}_o^\top\|_2$. This satisfies the second assumption.

On the other hand, for classification and next-token prediction tasks, the output layer is

$$\hat{\mathbf{y}} = \text{Softmax}(\boldsymbol{\psi}_o \mathbf{x}(T)) = \frac{e^{\boldsymbol{\psi}_o \mathbf{x}(T)}}{\mathbf{1}_c^\top e^{\boldsymbol{\psi}_o \mathbf{x}(T)}}, \tag{34}$$

and the loss function is the cross-entropy loss

$$\text{CrossEntropy}(\hat{\mathbf{y}}, \mathbf{y}) = -\mathbf{y}^\top \log(\hat{\mathbf{y}}). \tag{35}$$

Here, $c$ is the number of classes, $\mathbf{1}_c \in \mathbb{R}^c$ is a vector of all ones, and $\hat{\mathbf{y}}, \mathbf{y} \in \Delta^{c-1}$, which lie in the $(c-1)$-dimensional simplex, represent the model output and target output, respectively. The loss function is the log-sum-exp function plus a linear term [46]

$$
\begin{aligned}
G(\mathbf{x}(T), \mathbf{y}) = \mathrm{CrossEntropy}(\mathrm{Softmax}(\boldsymbol{\psi}_o \mathbf{x}(T)), \mathbf{y}) &= -\mathbf{y}^\top \log \frac{e^{\boldsymbol{\psi}_o \mathbf{x}(T)}}{\mathbf{1}_c^\top e^{\boldsymbol{\psi}_o \mathbf{x}(T)}} \\
&= -\mathbf{y}^\top \boldsymbol{\psi}_o \mathbf{x}(T) + (\mathbf{y}^\top \mathbf{1}_c) \log \left( \mathbf{1}_c^\top e^{\boldsymbol{\psi}_o \mathbf{x}(T)} \right) \\
&= \underbrace{-\mathbf{y}^\top \boldsymbol{\psi}_o \mathbf{x}(T)}_{\text{linear term}} + \underbrace{\log \left( \mathbf{1}_c^\top e^{\boldsymbol{\psi}_o \mathbf{x}(T)} \right)}_{\text{log-sum-exp}}.
\end{aligned}
$$

Here, in the last step, we used the fact that $(\mathbf{y}^\top \mathbf{1}_c) = 1$, because $\mathbf{y} \in \Delta^{c-1}$. Since the log-sum-exp function is convex and smooth (hence twice continuously differentiable) [46], so is $G$. Moreover, $G$ is proper because its value is always nonnegative and has nonempty effective domain. This shows that the first assumption is satisfied. Next, the gradient of $G$ is

$$
\begin{aligned}
\nabla G(\mathbf{x}(T), \mathbf{y}) &= -\boldsymbol{\psi}_o^\top \mathbf{y} + \boldsymbol{\psi}_o^\top \mathrm{diag}(e^{\boldsymbol{\psi}_o \mathbf{x}(T)}) \mathbf{1}_c \frac{1}{\mathbf{1}_c^\top e^{\boldsymbol{\psi}_o \mathbf{x}(T)}} \\
&= -\boldsymbol{\psi}_o^\top \mathbf{y} + \boldsymbol{\psi}_o^\top \frac{e^{\boldsymbol{\psi}_o \mathbf{x}(T)}}{\mathbf{1}_c^\top e^{\boldsymbol{\psi}_o \mathbf{x}(T)}} \\
&= -\boldsymbol{\psi}_o^\top \mathbf{y} + \boldsymbol{\psi}_o^\top \mathrm{Softmax}(\boldsymbol{\psi}_o \mathbf{x}(T)).
\end{aligned}
$$

Here $\mathrm{diag}(\mathbf{z})$ denotes a diagonal matrix with diagonal entries equal to $\mathbf{z}$. It is straightforward to verify that $\nabla G$ is Lipschitz continuous with respect to its second arguement with Lipschitz constant $L = \|\boldsymbol{\psi}_o^\top\|_2$. Note that the softmax function $\mathrm{Softmax}(\mathbf{z}) = \frac{e^{\mathbf{z}}}{\mathbf{1}_c^\top e^{\mathbf{z}}}$ is Lipschitz continuous with a Lipschitz constant 1 [29]. For any $\mathbf{x}_1(T), \mathbf{x}_2(T)$,

$$
\begin{aligned}
&\|\nabla G(\mathbf{x}_1(T), \mathbf{y}) - \nabla G(\mathbf{x}_2(T), \mathbf{y})\|_2 \\
&= \|-\boldsymbol{\psi}_o^\top \mathbf{y} + \boldsymbol{\psi}_o^\top \mathrm{softmax}(\boldsymbol{\psi}_o \mathbf{x}_1(T)) - (-\boldsymbol{\psi}_o^\top \mathbf{y} + \boldsymbol{\psi}_o^\top \mathrm{softmax}(\boldsymbol{\psi}_o \mathbf{x}_2(T)))\|_2 \\
&= \|\boldsymbol{\psi}_o^\top (\mathrm{softmax}(\boldsymbol{\psi}_o \mathbf{x}_1(T)) - \mathrm{softmax}(\boldsymbol{\psi}_o \mathbf{x}_2(T))) \|_2 \\
&\leq \|\boldsymbol{\psi}_o^\top\|_2 \|\mathrm{softmax}(\boldsymbol{\psi}_o \mathbf{x}_1(T)) - \mathrm{softmax}(\boldsymbol{\psi}_o \mathbf{x}_2(T))\|_2 \\
&\leq \|\boldsymbol{\psi}_o^\top\|_2 \|\boldsymbol{\psi}_o \mathbf{x}_1(T) - \boldsymbol{\psi}_o \mathbf{x}_2(T)\|_2 \quad \text{(since softmax has Lipschitz constant 1)} \\
&\leq \|\boldsymbol{\psi}_o^\top\|_2 \|\boldsymbol{\psi}_o\|_2 \|\mathbf{x}_1(T) - \mathbf{x}_2(T)\|_2 \\
&= L^2 \|\mathbf{x}_1(T) - \mathbf{x}_2(T)\|_2.
\end{aligned}
$$

Therefore, $\nabla G$ is Lipschitz continuous in both arguments, and the second assumption is satisfied.

Our derivation also reveals that for regression, classification and next-token prediction tasks, $\nabla G$ is Lipschitz continuous in each of its two arguments, with the same Lipschitz constants in all cases. We summarize this finding in the following lemma.

**Lemma 1.** $\nabla G(\cdot, \cdot)$ *is Lipschitz continuous in each of its two arguments, with Lipschitz constants $L^2$ and $L$, respectively, where $L$ is the 2-operator norm of the output layer weights $\boldsymbol{\psi}_o$.*

### F.2 Mean Field Control Formulation

We first recall the non-parametric formulation of the training problem (10):

$$
\begin{aligned}
\min_f \quad & \mathbb{E}_{(\mathbf{X}_0, \mathbf{y})} \left\{ G(\mathbf{X}(T), \mathbf{y}) + \frac{\lambda}{2} \int_0^T \|f(\mathbf{X}(s), s)\|_F^2 \, ds \right\}, \\
\text{subject to} \quad & \frac{d\mathbf{X}(t)}{dt} = f(\mathbf{X}(t), t), \quad \text{for} \quad t \in [0, T], \\
\text{and} \quad & \mathbf{X}(0) = \mathbf{X}_0.
\end{aligned}
\tag{36}
$$

Here, for clarity of exposition, we denote $s$ as the dummy variable for time integration.

The formulation (36) can be rewritten into a mean field control problem given by

$$\min_{f,\rho} \quad \mathcal{G}(\rho(\cdot,\cdot,T)) + \frac{\lambda}{2} \int_0^T \int_{\mathbb{R}^{d \times n}} \|f(\mathbf{X},s)\|_F^2 \rho(\mathbf{X},\mathbf{y},s) \, d\mathbf{X} ds, \tag{37}$$

$$\text{subject to} \quad \partial_t \rho(\mathbf{X},\mathbf{y},t) + \nabla \cdot (\rho(\mathbf{X},\mathbf{y},t) f(\mathbf{X},t)) = 0, \quad \text{for} \quad t \in [0,T], \tag{38}$$

$$\text{and} \quad \rho(\mathbf{X},\mathbf{y},0) = \rho_0(\mathbf{X},\mathbf{y}), \quad \text{for} \quad \mathbf{X} \in \mathbb{R}^{d \times n}. \tag{39}$$

Here, a mean field perspective is adopted by modeling the evolution of the density function $\rho : \mathbb{R}^{d \times n} \times \mathbb{R}^c \times [0,T] \to \mathbb{R}_{\geq 0}$, which characterizes the distribution of the hidden state for $t \in [0,T]$ given the target output $\mathbf{y}$. In particular, the probability density of the hidden state evolves continuously over time under the velocity field $f$, as governed by the continuity equation (38). The initial density $\rho_0 : \mathbb{R}^{d \times n} \times \mathbb{R}^c \to \mathbb{R}_{\geq 0}$ defines the joint distribution of the target output $\mathbf{y}$ and the hidden states at $t = 0$ (that is, the embedded input (2)). The first term of the objective is given by

$$\mathcal{G}(\rho(\cdot,\cdot,T)) = \mathbb{E}_{(\mathbf{X},\mathbf{y}) \sim \rho(\cdot,\cdot,T)} G(\mathbf{X},\mathbf{y}) \,. \tag{40}$$

### F.3  Optimal Control Theory

We review optimal control theory that will be used in our analysis.

We first recall the definition of the variational derivative of a functional. For a test function $w \in L^2(\mathbb{R}^{dn+c})$, the variational derivative $\frac{\delta \mathcal{G}}{\delta \rho}$ of $\mathcal{G}$ with respect to $\rho$ is defined as

$$\lim_{h \to 0} \frac{\mathcal{G}(\rho + hw) - \mathcal{G}(\rho)}{h} = \int_{\mathbb{R}^c} \int_{\mathbb{R}^{d \times n}} \frac{\delta \mathcal{G}(\rho)}{\delta \rho}(\mathbf{X},\mathbf{y}) \, w(\mathbf{X},\mathbf{y}) \, d\mathbf{X} d\mathbf{y}. \tag{41}$$

Evaluating (41) using the definition of $\mathcal{G}$ (40), we have

$$\frac{\delta \mathcal{G}(\rho(\cdot,T))}{\delta \rho}(\mathbf{X},\mathbf{y}) = G(\mathbf{X},\mathbf{y}). \tag{42}$$

Consider the mean field control problem (37)-(39). Under Lagrangian formulation, for a hidden state at location $\mathbf{X}$ and time $t$ with target output $\mathbf{y}$, its potential function $\Phi_{\mathbf{y}} : \mathbb{R}^{d \times n} \times [0,T] \to \mathbb{R}$ is given by [51]

$$\begin{aligned}
\Phi_{\mathbf{y}}(\mathbf{X},t) &= \inf_{f,\rho} \left\{ \frac{\delta \mathcal{G}(\rho(\cdot,\cdot,T))}{\delta \rho}(\mathbf{X}(T),\mathbf{y}) + \frac{\lambda}{2} \int_t^T \|f(\mathbf{X}(s),s)\|_F^2 \, ds \right\} \\
&= \inf_f \left\{ G(\mathbf{X}(T),\mathbf{y}) + \frac{\lambda}{2} \int_t^T \|f(\mathbf{X}(s),s)\|_F^2 \, ds \right\}.
\end{aligned} \tag{43}$$

Here, we used (42) in the last step.

Under standard regularity assumptions [25][Section 10.3.1], the potential function is bounded and Lipschitz continuous in both arguments [25]. Moreover, when $\lambda > 0$, the potential function is the unique solution to the Hamilton-Jacobi-Bellman (HJB) partial differentiation equation (PDE)

$$-\partial_t \Phi_{\mathbf{y}}(\mathbf{X},t) + H(\nabla \Phi_{\mathbf{y}}(\mathbf{X},t)) = 0, \tag{44}$$

$$\Phi_{\mathbf{y}}(\mathbf{X},T) = \frac{\delta \mathcal{G}(\rho(\cdot,T))}{\delta \rho}(\mathbf{X},\mathbf{y}), \tag{45}$$

where the gradient $\nabla_{\mathbf{y}} \Phi(\mathbf{X},t)$ is taken with respect to the spatial variable. And the Hamiltonian $H : \mathbb{R}^{d \times n} \to \mathbb{R}$ is given by

$$H(\mathbf{P}) = \sup_f \left\{ -\langle \mathbf{P},f \rangle - \frac{\lambda}{2} \|f\|_F^2 \right\}, \tag{46}$$

where $\mathbf{P}$ is the costate variable and $\langle \cdot,\cdot \rangle$ denotes the Frobenius inner product. When $\lambda > 0$, the objective is strictly concave, and thus the supremum is uniquely attained when $f = -\frac{1}{\lambda}\mathbf{P}$. The Hamiltonian becomes

$$H(\mathbf{P}) = \frac{1}{2\lambda} \|\mathbf{P}\|_F^2. \tag{47}$$

Substituting (42) into (45), and (47) into (44), the HJB equation becomes

$$-\partial_t \Phi_\mathbf{y}(\mathbf{X}, t) + \frac{1}{2\lambda} \|\nabla \Phi_\mathbf{y}(\mathbf{X}, t)\|_F^2 = 0, \tag{48}$$

$$\Phi_\mathbf{y}(\mathbf{X}, T) = G(\mathbf{X}, \mathbf{y}). \tag{49}$$

Moreover, the derivation of the HJB equation reveals that the optimal velocity field $f^*$ of the mean field control problem (37) to (39) attains the supremum in the Hamiltonian when $\mathbf{P} = \nabla \Phi(\mathbf{X}, t)$, thus we have

$$f^*(\mathbf{X}, t) = -\frac{1}{\lambda} \nabla \Phi_\mathbf{y}(\mathbf{X}, t). \tag{50}$$

### F.4 Well-posedness of Training Problem

We prove Theorem 1. We first restate the theorem.

**Theorem 1.** *There exists a unique solution to the optimization problem* (10) *if and only if* $\lambda > 0$.

*Proof.* When $\lambda = 0$, the Hamtiltonian (46) is degenerate and unbounded above, and the optimal velocity can have an arbitrarily large magnitude. This implies that the HJB PDE is not well-defined. As such, (37)-(39) reduces to a degenerate mean field control problem that has infinitely many solutions.

We note that $\lambda > 0$ if and only if the Hamiltonian (46) is well-defined and admits a unique maximizer. Furthermore, the HJB PDE (44)-(45) is well-defined if and only if the Hamiltonian is well-defined. In this case, the potential function is the unique solution to the HJB PDE [25]. Thus, the optimal velocity, which is given by the gradient of the potential function (50), is also unique.

$$\square$$

### F.5 Highly Regular Trajectory

Next, we show that the unique velocity guaranteed by Theorem 1 when $\lambda > 0$ is highly regular. In particular, the OT-Transformer produces exactly straight trajectories.

**Proposition 1.** *When* $\lambda > 0$, *under the optimal velocity* $f^*$, *each hidden state* $\mathbf{X}(t)$ *travels along a straight trajectory at a constant speed.*

*Proof.* We prove that the optimal velocity $f^*(\mathbf{X}(t), t)$ remains constant for $t \in [0, T]$, which induces straight trajectories and constant speed.

From the Pontryagin Maximum Principle [67], we have that

$$\frac{d\mathbf{X}(t)}{dt} = -\nabla_\mathbf{P} H(\mathbf{P}(t)), \tag{51}$$

$$\frac{d\mathbf{P}(t)}{dt} = \nabla_\mathbf{X} H(\mathbf{P}(t)) = 0, \tag{52}$$

$$\mathbf{P}(T) = \nabla G(\mathbf{X}(T), \mathbf{y}). \tag{53}$$

Here $\mathbf{X}(t)$ denote the trajectory induced by $f^*$, and $\mathbf{P}(t)$ is the corresponding costate variable. The gradient $\nabla G(\mathbf{X}(T), \mathbf{y})$ is taken with respect to the first argument. In the second step of (52), we used the fact that $H(\mathbf{P}(t))$ is independent of $\mathbf{X}$.

From (52), we see that $\mathbf{P}(t)$ remains constant for all $t \in [0, T]$. Thus, we have

$$\mathbf{P}(t) = \mathbf{P}(T) = \nabla G(\mathbf{X}(T), \mathbf{y}) \quad \text{for all} \quad t \in [0, T], \tag{54}$$

where we applied (53) in the final step. Differentiating the formula of the Hamiltonian (47), we have

$$\nabla_\mathbf{P} H(\mathbf{P}) = \frac{1}{\lambda} \mathbf{P}. \tag{55}$$

Substituting (54) and (55) into (51), we obtain

$$\frac{d\mathbf{X}(t)}{dt} = -\frac{1}{\lambda} \mathbf{P}(t) = -\frac{1}{\lambda} \mathbf{P}(T) = -\frac{1}{\lambda} \nabla G(\mathbf{X}(T), \mathbf{y}), \tag{56}$$

for all $t \in [0, T]$. This shows that the velocity for each hidden state is constant for all $t \in [0, T]$. This implies each hidden state travels along a straight line at constant speed. $\quad \square$

The regularized trajectories present a stark contrast to those of the unregularized case. Thanks to such highly regular trajectories, the numerical integration of the dynamical system (7) is particularly easy to perform. Practically, it requires fewer time steps to achieve accurate integration. Furthermore, since there are no fluctuations or abnormal magnitude changes in the velocity (as is for the unregularized case), numerical stability during integration, and consequently during training, is significantly improved. This effect is evident in our experiments, where the unregularized training exhibits instability, while the regularized training does not encounter such issues.

We note that the straight trajectories and constant speeds are not immediately apparent and are not guaranteed to hold in general. On one hand, the training problem is a *relaxed* optimal transport problem (see Appendix E for details), which does not guarantee such regularity in general. On the other hand, this regularity is guaranteed only under our assumptions; see Appendix F.1. The proof applies the Pontryagin Maximum Principle [67] and leverages the regularity assumptions on the loss function $G$; see Appendix F.5 for details.

### F.6 Regularity of Potential Function

Next, we show that for each given target output $\mathbf{y}$, the hidden states trajectories never intersect.

**Proposition 2.** *For two hidden states with different initial conditions $\mathbf{X}_0$ and $\tilde{\mathbf{X}}_0$ and the same target output $\mathbf{y}$, their trajectories never intersect.*

This proposition implies that the solution to the HJB PDE (48)-(49) does not develop shocks. In other words, the gradient of the potential function $\nabla \Phi_{\mathbf{y}}$ is continuous in space.

*Proof.* Integrating (56) from $t$ to $T$, the trajectory of a hidden state $\mathbf{X}(t)$ is given by

$$\mathbf{X}(t) = \mathbf{X}(T) + \frac{(T-t)}{\lambda}\nabla G(\mathbf{X}(T), \mathbf{y}), \tag{57}$$

for all $t \in [0, T]$. We first show that the trajectories corresponding to two different *terminal states* $\mathbf{X}_T$ and $\tilde{\mathbf{X}}_T$ at $t = T$ and the same target output $\mathbf{y}$ cannot intersect. That is,

$$\mathbf{X}_T + \frac{(T-t)}{\lambda}\nabla G(\mathbf{X}_T, \mathbf{y}) = \tilde{\mathbf{X}}_T + \frac{(T-t)}{\lambda}\nabla G(\tilde{\mathbf{X}}_T, \mathbf{y}) \tag{58}$$

cannot hold for any $t \in [0, T]$ if $\mathbf{X}_T \neq \tilde{\mathbf{X}}_T$. To this end, consider the function

$$F_t(\mathbf{X}) = \frac{1}{2}\|\mathbf{X}\|_F^2 + \frac{T-t}{\lambda}G(\mathbf{X}, \mathbf{y}), \tag{59}$$

which is strictly convex with respect to $\mathbf{X}$ for all $t \in [0, T]$, as $G$ is convex with respect to $\mathbf{X}$ by assumption. Note that its gradient with respect to $\mathbf{X}$ is

$$\nabla F_t(\mathbf{X}) = \mathbf{X} + \frac{T-t}{\lambda}\nabla G(\mathbf{X}, \mathbf{y}), \tag{60}$$

which coincides the trajectory formulation in (57). By the strict convexity of $F_t$, its gradient $\nabla F_t$ is injective. Thus, (58) does not hold for any $\mathbf{X}_T \neq \tilde{\mathbf{X}}_T$.

Next, we argue that for two hidden states with different initial conditions, the corresponding terminal states at $t = T$ are different. Combining (57) and (60), at $t = 0$ we have

$$\mathbf{X}(0) = \nabla F_0(\mathbf{X}(T)). \tag{61}$$

Since $\nabla F_0$ is the gradient of a strictly convex function, its inverse $(\nabla F_0)^{-1}$, defined by

$$\mathbf{X}(T) = (\nabla F_0)^{-1}(\mathbf{X}(0)), \tag{62}$$

exists and is also the gradient of a strictly convex function (Lemma 2 of Appendix G). Using the same injectivity arguments on (62), we have that for two hidden states with different initial conditions $\mathbf{X}_0$ and $\tilde{\mathbf{X}}_0$, their terminal states $\mathbf{X}_T$ and $\tilde{\mathbf{X}}_T$ are different. And we have established in the previous paragraph that trajectories with different terminal states $\mathbf{X}_T$ and $\tilde{\mathbf{X}}_T$ cannot intersect. Thus, we have proven that trajectories with different initial states cannot intersect. $\square$

### F.7 Stable Forward Propagation

**Theorem 2.** *Assume that the OT regularization hyperparameter is such that $\lambda > TL^2$, where $T$ is the terminal time and $L$ is the 2-operator norm of the output layer weights (5). For embedded input-output pairs $(\mathbf{X}_1(0), \mathbf{y}_1)$ and $(\mathbf{X}_2(0), \mathbf{y}_2)$, the corresponding model outputs $\tilde{\mathbf{y}}_1$ and $\tilde{\mathbf{y}}_2$ satisfy*

$$\|\tilde{\mathbf{y}}_1 - \tilde{\mathbf{y}}_2\|_2 \leq \left(1 - \frac{TL^2}{\lambda}\right)^{-1} \left(L\|\mathbf{X}_1(0) - \mathbf{X}_2(0)\|_F + \frac{TL^2}{\lambda}\|\mathbf{y}_1 - \mathbf{y}_2\|_2\right). \tag{63}$$

*Proof.* By Proposition 1 and Proposition 2, the optimal velocity $f^*$ remains constant along the trajectory $\mathbf{X}(t)$. Thus, by (56), the trajectory is given by

$$\frac{d\mathbf{X}(t)}{dt} = f^*(\mathbf{X}(t), t) = -\frac{1}{\lambda}\nabla G(\mathbf{X}(T), \mathbf{y}). \tag{64}$$

Integrating (64) from $t = 0$ to $t = T$, we obtain

$$\mathbf{X}(T) = \mathbf{X}(0) - \frac{T}{\lambda}\nabla G(\mathbf{X}(T), \mathbf{y}). \tag{65}$$

We now bound the difference between the terminal states $\mathbf{X}_1(T)$ and $\mathbf{X}_2(T)$ as follows

$$\|\mathbf{X}_1(T) - \mathbf{X}_2(T)\|_F = \left\|\mathbf{X}_1(0) - \frac{T}{\lambda}\nabla G(\mathbf{X}_1(T), \mathbf{y}_1) - \left(\mathbf{X}_2(0) - \frac{T}{\lambda}\nabla G(\mathbf{X}_2(T), \mathbf{y}_2)\right)\right\|_F \tag{66}$$

$$\leq \|\mathbf{X}_1(0) - \mathbf{X}_2(0)\|_F + \frac{T}{\lambda}\|\nabla G(\mathbf{X}_1(T), \mathbf{y}_1) - \nabla G(\mathbf{X}_2(T), \mathbf{y}_2)\|_F \tag{67}$$

$$= \|\mathbf{X}_1(0) - \mathbf{X}_2(0)\|_F$$
$$+ \frac{T}{\lambda}\|\nabla G(\mathbf{X}_1(T), \mathbf{y}_1) - \nabla G(\mathbf{X}_2(T), \mathbf{y}_1) \tag{68}$$
$$+ \nabla G(\mathbf{X}_2(T), \mathbf{y}_1) - \nabla G(\mathbf{X}_2(T), \mathbf{y}_2)\|_F$$

$$\leq \|\mathbf{X}_1(0) - \mathbf{X}_2(0)\|_F$$
$$+ \frac{T}{\lambda}\|\nabla G(\mathbf{X}_1(T), \mathbf{y}_1) - \nabla G(\mathbf{X}_2(T), \mathbf{y}_1)\|_F \tag{69}$$
$$+ \frac{T}{\lambda}\|\nabla G(\mathbf{X}_2(T), \mathbf{y}_1) - \nabla G(\mathbf{X}_2(T), \mathbf{y}_2)\|_F$$

$$\leq \|\mathbf{X}_1(0) - \mathbf{X}_2(0)\|_F$$
$$+ \frac{T}{\lambda}L^2\|\mathbf{X}_1(T) - \mathbf{X}_2(T)\|_F \tag{70}$$
$$+ \frac{T}{\lambda}L\|\mathbf{y}_1 - \mathbf{y}_2\|_2, \quad \text{by Lemma 1.}$$

By the assumption that $\lambda > TL^2$, we have $\frac{TL^2}{\lambda} < 1$. Thus, (70) becomes

$$\left(1 - \frac{TL^2}{\lambda}\right)\|\mathbf{X}_1(T) - \mathbf{X}_2(T)\|_F \leq \|\mathbf{X}_1(0) - \mathbf{X}_2(0)\|_F + \frac{TL}{\lambda}\|\mathbf{y}_1 - \mathbf{y}_2\|_2 \tag{71}$$

$$\|\mathbf{X}_1(T) - \mathbf{X}_2(T)\|_F \leq \left(1 - \frac{TL^2}{\lambda}\right)^{-1}\left(\|\mathbf{X}_1(0) - \mathbf{X}_2(0)\|_F + \frac{TL}{\lambda}\|\mathbf{y}_1 - \mathbf{y}_2\|_2\right). \tag{72}$$

Recall that for the case of regression, the model output (5) is given by

$$\tilde{\mathbf{y}} = g_o(\mathbf{X}(T); \boldsymbol{\psi}_o) = \boldsymbol{\psi}_o\text{vec}(\mathbf{X}(T)), \tag{73}$$

which is obviously Lipschitz continuous with respect to $\text{vec}(\mathbf{X}(T))$ with a Lipschitz constant $L = \|\boldsymbol{\psi}_o\|_F$. For the case of classification and next-token generation, the model output (5) is given by

$$\tilde{\mathbf{y}} = g_o(\mathbf{X}(T); \boldsymbol{\psi}_o) = \text{Softmax}(\boldsymbol{\psi}_o\text{vec}(\mathbf{X}(T))) = \frac{e^{\boldsymbol{\psi}_o\text{vec}(\mathbf{X}(T))}}{\mathbf{1}_c^\top e^{\boldsymbol{\psi}_o\text{vec}(\mathbf{X}(T))}}. \tag{74}$$

Since the softmax function is Lipschitz continuous with Lipschitz constant 1 [29], the model output $\tilde{y}$ is also Lipschitz continuous with respect to $\text{vec}(\mathbf{X}(T))$ with the same Lipschitz constant $L = \|\boldsymbol{\psi}_o\|_F$.

Thus, the corresponding model outputs $\tilde{\mathbf{y}}_1$ and $\tilde{\mathbf{y}}_2$ satisfy

$$
\begin{aligned}
\|\tilde{\mathbf{y}}_1 - \tilde{\mathbf{y}}_2\|_2 &\leq L\|\text{vec}(\mathbf{X}_1(T)) - \text{vec}(\mathbf{X}_2(T))\|_2 \\
&= L\|\mathbf{X}_1(T) - \mathbf{X}_2(T)\|_F \\
&\leq \left(1 - \frac{TL^2}{\lambda}\right)^{-1}\left(L\|\mathbf{X}_1(0) - \mathbf{X}_2(0)\|_F + \frac{TL^2}{\lambda}\|\mathbf{y}_1 - \mathbf{y}_2\|_2\right), \quad \text{by (72)}.
\end{aligned}
$$

$\square$

## G  Auxiliary Lemma

We state and prove an auxiliary lemma, which is used in the proof of Proposition 2 in Appendix F. The proof modifies the argument from [44, Proposition 1].

**Lemma 2.** *Let $\mathcal{D} \subseteq \mathbb{R}^k$ be open, $H : \mathcal{D} \to \mathbb{R}$ be a strictly convex and continuously differentiable function and $h$ be its gradient. Then $h^{-1}$ exists and is the gradient of a strictly convex function.*

*Proof.* The strict convexity and continuous differentiability of $H$ implies the existence of $h^{-1}$ and that $\nabla h(\mathbf{x})$ is symmetric positive definite (SPD) for all $\mathbf{x} \in \mathcal{D}$. Since $h$ is continuously differentiable and $\nabla h(\mathbf{x})$ is SPD for all $\mathbf{x} \in \mathcal{D}$, by the Inverse Function Theorem [72, Theorem 9.24], $h^{-1}$ is also continuously differentiable and

$$
\nabla h^{-1}(h(\mathbf{x}))\nabla h(\mathbf{x}) = \mathbf{I}_k \quad \text{for all} \quad \mathbf{x} \in \mathcal{D}.
$$

Here $\mathbf{I}_k$ is the $k \times k$ identity matrix. This implies $\nabla h^{-1}(\mathbf{y})$ is also SPD for all $\mathbf{y} \in h(\mathcal{D})$. Thus, by the symmetry of $\nabla h^{-1}(\mathbf{y})$, we have

$$
\frac{\partial [h^{-1}]_i}{\partial y_j} = \frac{\partial [h^{-1}]_j}{\partial y_i} \quad \text{for all } i, j = 1, 2, ..., k. \tag{75}
$$

Here, $y_j$ denotes the $j$th entry of $\mathbf{y}$. Let $\psi(\mathbf{y}) = \sum_{j=1}^k y_j \int_0^1 [h^{-1}]_j(t\mathbf{y})dt$. Consider its partial derivative

$$
\frac{\partial \psi}{\partial y_i}(\mathbf{y}) = \int_0^1 [h^{-1}]_i(t\mathbf{y})dt + \int_0^1 \sum_{j=1}^k y_j t \frac{\partial [h^{-1}]_j}{\partial y_i}(t\mathbf{y})dt
$$

using (75), we get

$$
= \int_0^1 [h^{-1}]_i(t\mathbf{y})dt + \int_0^1 \sum_{j=1}^k y_j t \frac{\partial [h^{-1}]_i}{\partial y_j}(t\mathbf{y})dt
$$

applying the chain rule $\frac{\partial}{\partial t}\left([h^{-1}]_i(t\mathbf{y})\right) = \sum_{j=1}^k y_j \frac{\partial [h^{-1}]_i}{\partial y_j}(t\mathbf{y})$, we obtain

$$
= \int_0^1 [h^{-1}]_i(t\mathbf{y})dt + \int_0^1 t \frac{\partial}{\partial t}\left([h^{-1}]_i(t\mathbf{y})\right)dt
$$

performing integration by parts, we have

$$
\begin{aligned}
&= \cancel{\int_0^1 [h^{-1}]_i(t\mathbf{y})dt} + t[h^{-1}]_i(t\mathbf{y})\big|_{t=0}^{t=1} - \cancel{\int_0^1 [h^{-1}]_i(t\mathbf{y})dt} \\
&= [h^{-1}]_i(\mathbf{y}), \quad \text{for} \quad i = 1, 2, ..., k.
\end{aligned}
$$

Therefore, $\nabla \psi = h^{-1}$. Moreover $\psi$ is strictly convex because its Hessian $\nabla h^{-1}(\mathbf{y})$ is SPD for all $\mathbf{y} \in h(\mathcal{D})$. Therefore, $h^{-1}$ is the gradient of the strictly convex function $\psi$. $\square$

# H  Derivations for Robustness and Generalization

In this section, we provide detailed derivations for the robustness and generalization properties discussed in Section 1 and Section 4.

Recall the stability in forward propagation inequality

$$\|\tilde{\mathbf{y}}_1 - \tilde{\mathbf{y}}_2\|_2 \le C\|\mathbf{X}_1(0) - \mathbf{X}_2(0)\|_F + C'\|\mathbf{y}_1 - \mathbf{y}_2\|_2, \tag{76}$$

where $C = \left(1 - \frac{TL^2}{\lambda}\right)^{-1} L$ and $C' = \left(1 - \frac{TL^2}{\lambda}\right)^{-1} \frac{TL^2}{\lambda}$.

## H.1  Distributional robustness

We first provide a proof of Theorem 3, the distributional robustness inequality

$$W_p(\mathbf{T}_{\#}\mu, \mathbf{T}_{\#}\nu) \le \tilde{C}W_p(\mu, \nu), \quad \text{for any } p \ge 1, \tag{77}$$

where $\mathbf{T}_{\#}$ denotes the pushforward operator under the trained Transformer.

*Proof.* Define $\mathbf{x}_1 = \text{vec}(\mathbf{X}_1(0))$ and $\mathbf{x}_2 = \text{vec}(\mathbf{X}_2(0))$, and note that $\|\mathbf{X}_1(0) - \mathbf{X}_2(0)\|_F = \|\mathbf{x}_1 - \mathbf{x}_2\|_2$. Suppose $\mathbf{x}_1 \sim \mu$ and $\mathbf{x}_2 \sim \nu$. Holding the output data to be the same, the stable forward propagation inequality gives us $\|\mathbf{T}(\mathbf{x}_1) - \mathbf{T}(\mathbf{x}_2)\|_2 \le C\|\mathbf{x}_1 - \mathbf{x}_2\|_2$. By equivalence of $p$-norms we have that there exists a constant $\hat{C}(p) > 0$ such that

$$\|\mathbf{T}(\mathbf{x}_1) - \mathbf{T}(\mathbf{x}_2)\|_p \le \hat{C}(p)L(1 - TL^2\lambda^{-1})^{-1}\|\mathbf{x}_1 - \mathbf{x}_2\|_p. \tag{78}$$

For a fixed $p$, let $\pi \in \Gamma(\mu, \nu)$ to be the optimal coupling between $\mu$ and $\nu$. Then the pushforward of $\pi$ under the map $(\mathbf{T} \otimes \mathbf{T})(\mathbf{x}_1, \mathbf{x}_2) = (\mathbf{T}(\mathbf{x}_1), \mathbf{T}(\mathbf{x}_2))$ is a suboptimal coupling between measures $\mathbf{T}_{\sharp}\mu$ and $\mathbf{T}_{\sharp}\nu$. Therefore, we have

$$\begin{aligned}
W_p^p(\mathbf{T}_{\sharp}\mu, \mathbf{T}_{\sharp}\nu) &\le \int \|\mathbf{y}_1 - \mathbf{y}_2\|_p^p d(\mathbf{T} \otimes \mathbf{T})_{\sharp}\pi(\mathbf{y}_1, \mathbf{y}_2) \\
&= \int \|\mathbf{T}(\mathbf{x}_1) - \mathbf{T}(\mathbf{x}_2)\|_p^p d\pi(\mathbf{x}_1, \mathbf{x}_2) \\
&\le \hat{C}(p)^p L^p (1 - TL^2\lambda^{-1})^{-p} W_p^p(\mu, \nu).
\end{aligned}$$

Taking the $p$-th root on each side, we obtain the desired result. $\qquad\square$

## H.2  Derivation of Robustness Under Sampling Error or Data Perturbation

Note that the data used for training always come from the underlying true distribution in theory; however, sampling error and perturbation error may occur when generating the training dataset. We show here that the forward model remains accurate when such errors are small.

To derive this property, we first specify the variables. Let $\mu$ and $\tilde{\mu}$ denote the training distributions for the input and output data, and $\nu$ and $\tilde{\nu}$ the true distributions for the input and output data, respectively. Denote $\mathbf{T}_{\#}$ as the pushforward operator under the trained Transformer.

Our goal is to show that $W_p(\mathbf{T}_{\#}\nu, \tilde{\nu})$ remains bounded. That is, the pushforward (induced by the trained Transformer) of the true distribution for input data closely approximates the true distribution of output data.

Applying triangle inequality to (77), we obtain

$$-W_p(\mathbf{T}_{\#}\mu, \tilde{\mu}) - W_p(\tilde{\mu}, \tilde{\nu}) + W_p(\mathbf{T}_{\#}\nu, \tilde{\nu}) \le W_p(\mathbf{T}_{\#}\mu, \mathbf{T}_{\#}\nu) \le \tilde{C}W_p(\mu, \nu).$$

The use of Wasserstein metrics is crucial in this context. The derivation above relies on the fact that the Wasserstein distance $W_p$ satisfies the triangle inequality—a property not shared by divergences such as the KL divergence, for which this argument would not hold. Thus, here we have

$$W_p(\mathbf{T}_{\#}\nu, \tilde{\nu}) \le \tilde{C} \underbrace{W_p(\mu, \nu)}_{\text{input distribution gap}} + \underbrace{W_p(\mathbf{T}_{\#}\mu, \tilde{\mu})}_{\approx\text{training loss}} + \underbrace{W_p(\tilde{\mu}, \tilde{\nu})}_{\text{output distribution gap}}. \tag{79}$$

Here, on the right hand side, the first term is the input distribution gap: it measures the difference between the training distribution and the true distribution of the input data. The third term measure the same for output data. We can bound both these terms using sample complexity results for Wasserstein metrics ($p \geq 1$) [27], see also (82) below. Lastly, the second term in (79) quantifies the difference between the training distribution for output data and the pushforward (induced by the trained Transformer) of the training distribution for input data. This term is an informative surrogate for the training loss. When the training loss decreases, this term will tend to decrease. And when the training loss is 0, this term will be 0 too.

Assuming the Transformer model is trained with sufficient accuracy, when the sampling error or perturbation error in the data is small, both the first and third terms will remain relatively small, showing that $W_p(\mathbf{T}_{\#}\nu, \tilde{\nu})$ is not only bounded but also small. Thus, Equation (79) indicates robustness with respect to the training dataset. The model's prediction will remain close to the underlying true distribution even in the presence of mild sampling error or data corruption.

### H.3 Out-of-Distribution Generalization

A great benefit of having Lipschitz functions is that we may obtain performance guarantees on the quality of the model learned from finite training samples is at making predictions on out-of-distribution samples. Suppose map $\mathbf{T}$ is trained on input samples $X_1, \ldots, X_N \sim \mu$ that form empirical measure $\hat{\mu}_N$. We use ideas from distributionally robust optimization (DRO) to produce performance bounds on out-of-sample performance and non-asymptotic generalization bounds [24, 75, 76].

Denote $\mathcal{W}_r(\hat{\mu}_N)$ to be the Wasserstein ball of radius $r$ around $\hat{\mu}_N$, i.e., $\mathcal{W}_r(\hat{\mu}_N) := \{\rho \in \mathcal{P}(\Omega) : W_1(\rho, \hat{\mu}_N) \leq r\}$. For simplicity, consider the setting where $\Omega$ is a bounded domain. Suppose we have out-of-distribution samples from distribution $\nu \in \mathcal{W}_r(\hat{\mu}_N)$, we wish to obtain bounds on the test error of the Transformer $\mathbf{T}$ on the out-of-distribution samples, i.e., $\mathbb{E}_\nu[G(\mathbf{T}(X), y)] = \mathbb{E}_{T_\sharp \nu}[G(Z, y)]$, where $Z = T(X)$ in terms of the training error $\mathbb{E}_{\hat{\mu}_N}[G(\mathbf{T}(X), y)] = \mathbb{E}_{T_\sharp \hat{\mu}_N}[G(Z, y)]$ where $G$ is the terminal condition of the training objective (8). As $\Omega$ is a bounded domain, $G$ is a Lipschitz function.

Observe that by Kantorovich duality, the 1-Wasserstein distance is given by the variational formula

$$W_1(\hat{\mu}_N, \nu) = \sup_{\varphi \in \mathrm{Lip}_1(\Omega)} \{\mathbb{E}_{\hat{\mu}_N}[\varphi(X)] - \mathbb{E}_\nu[\varphi(X)]\},$$

where $\mathrm{Lip}_1(\Omega)$ is the set of 1-Lipschitz functions on $\Omega$. Observe that with Theorem 3 we can obtain a bound on the expected test error with respect to the distribution $\nu \in \mathcal{W}_r(\hat{\mu}_N)$

$$\begin{aligned}
\mathbb{E}_{\mathbf{T}_\sharp \nu}[G(Z, y)] - \mathbb{E}_{\mathbf{T}_\sharp \hat{\mu}_N}[G(Z, y)] &\leq \|G\|_{\mathrm{Lip}} W_1(\mathbf{T}_\sharp \nu, \mathbf{T}_\sharp \hat{\mu}_N) \\
&\leq \|G\|_{\mathrm{Lip}} L(1 - \lambda^{-1} T L^2)^{-1} W_1(\nu, \hat{\mu}_N) \qquad (80) \\
&= \|G\|_{\mathrm{Lip}} L(1 - \lambda^{-1} T L^2)^{-1} r,
\end{aligned}$$

which implies that

$$\mathbb{E}_\nu[G(\mathbf{T}(X), y)] \leq \mathbb{E}_{\hat{\mu}_N}[G(\mathbf{T}(X), y)] + r\|G\|_{\mathrm{Lip}} L(1 - \lambda^{-1} T L^2)^{-1}. \qquad (81)$$

This bound is deterministic and holds for any $\nu \in \mathcal{W}_r(\hat{\mu}_N)$. If a statistical argument ensures this inclusion with high probability, then this becomes a generalization guarantee. In particular, if $\nu = \mu$, the distribution that generated the empirical distribution $\mu$, then $r$ is computable as a function of samples and we can obtain a non-asymptotic generalization bound.

### H.4 Generalization Bound via Concentration Inequalities

To turn the deterministic bound into a statistical generalization guarantee, first recall $\hat{\mu}_N$ to be the empirical measure constructed from the training data. Then $\mu \in \mathcal{W}_r(\hat{\mu}_N)$ with high probability–for a computable radius $r$–which is ensured by the concentration of the empirical measure in the Wasserstein distance. Indeed, under mild assumptions (e.g., bounded domains or sub-Gaussian tails), we have a sample complexity result [27]: there exists $C > 0$ such that with probability at least $1 - \delta$,

$$W_1(\mu, \hat{\mu}_N) \leq C N^{-1/d} + \sqrt{\frac{\log(1/\delta)}{N}} = r(N, \delta). \qquad (82)$$

Choosing the radius $r$ as a function of $\delta$ and the number of samples $N$ accordingly, we obtain, by combining (80) and (82), that with probability at least $1 - \delta$,

$$\mathbb{E}_\mu[G(\mathbf{T}(X), y)] \leq \mathbb{E}_{\hat{\mu}_N}[G(\mathbf{T}(X), y)] + r(\delta, N) \cdot \|G\|_{\mathrm{Lip}} L(1 - \lambda^{-1}TL^2)^{-1}. \tag{83}$$

This is a *non-asymptotic* generalization bound for Lipschitz losses in terms of the Wasserstein distance between $\mu$ and $\hat{\mu}_N$, linking distributional robustness of generalization performance. We emphasize this is only a crude application of DRO tools for analyzing Transformers that arise from optimal control tools. Finally, we note that Wasserstein distances are particularly well-suited for sample complexity analysis such as (82). In contrast to KL divergence, which requires absolute continuity between distributions, Wasserstein distances remain well-defined even when comparing an empirical measure $\hat{\mu}_N$—a discrete distribution—to a possibly continuous distribution $\mu$.

# I    Experimental Details and Results

We report the detailed experimental setups here. We adapted the code provided by [74, 95], maintaining the same default data processing setup, hyperparameters, and other experimental settings as used in their implementations. Our implementation is based on PyTorch [66] and experiments are conducted using NVIDIA A100 GPUs with 40GB of memory. Runtime varies by experiment. While Proposition 1 shows that the optimally trained model yields a straight hidden state trajectory with constant speed, in practice we use 8–20 integration steps and do not yet fully exploit this property to reduce runtime. In practice, fewer steps may suffice, which we leave for future work.

We also compared against N-ODE Transformer, an existing continuous-time Transformer formulation which is introduced in [5] and has been considered in other works. For details about the formulation and specific applications, see the discussion in Appendix A. In the reported results, we refer to N-ODE Transformer with and without transport cost as unregularized N-ODE Transformer and regularized N-ODE Transformer, respectively. The goal here is to demonstrate that our proposed approach is not only theoretically sound, but also excels in performance when compared against the baseline method and similar approaches.

**Point Cloud Classification.**    Given the ModelNet40 dataset, we experiment with the Set Transformer model [53]. It has an encoder-decoder architecture and is specifically designed to process unordered data, such as point clouds, ensuring that the output remains permutation invariant to its input. Following the setup of [74], we use a Set Transformer [53] with two Induced Self Attention Blocks (ISABs) in the encoder, where each ISAB contains two Transformer blocks, and with a Pooling by Multihead Attention (PMA) Module in the decoder. For each instance, we uniformly sample 5000 points from each element in the dataset. For the continuous-time models, we use the same architecture except that we put a fully-connected layer before the Transformer blocks so that the dimension is consistent for continuous-time dynamics. Also the hidden dimensions $d$ and $k$ of the ISABs are reduced from 256 to 200. This reduces the number of parameters for the ISABs by $24\%$. We use an Adam optimizer, with batch size 64, 200 training epochs, and learning rate of $1 \times 10^{-3}$. For the regularized N-ODE Transformer and OT-Transformer, the regularization hyperparameters are $\lambda = 0.1$ and $\lambda = 1$, respectively, as they provide the optimal performance in our tests. We use $T = 1$ and a total of 8 time steps for the numerical integration.

We perform the experiment over five random trials and report the best test accuracies in Table 6. The unregularized continuous-time models encountered gradient explosion, resulting in NaN outputs, and the issue persists even with slight regularization. We found that the models never suffered from gradient explosion with sufficient regularization, indicating that transport cost effectively stabilizes the training process. Hence, we only report the performance of the regularized models. The baseline Set Transformer obtains an average test accuracy of 87.4%. The regularized N-ODE Transformer achieves an accuracy of 87.5%, indicating negligible improvement over the vanilla model. Our OT-Transformer shows a sizable improvement and reports an average 89.9% test accuracy even with a smaller model. From the learning curves in Figure 2, we see that our model reports a lower data-fitting loss for training data compared to the vanilla model, despite the inclusion of a regularization term. This example also highlights the generalizability of our model: despite using an encoder-decoder architecture, it demonstrates clear advantages and noticeable performance improvements over the baseline.

Table 6: Number of parameters for the Transformer blocks, best and final test accuracies (with standard deviation) across five trials for the point cloud experiment.

| Method/Exp. | Para. Count | Best Test Accuracy | Final Test Accuracy |
|---|---|---|---|
| Baseline | 0.86M | $87.4\% \pm 0.45\%$ | $86.6\% \pm 0.67\%$ |
| Reg. N-ODE Trans. | 0.65M | $87.5\% \pm 0.51\%$ | $86.7\% \pm 0.43\%$ |
| OT-Trans. (Ours) | 0.65M | $\mathbf{89.9\% \pm 0.42\%}$ | $\mathbf{89.3\% \pm 0.69\%}$ |

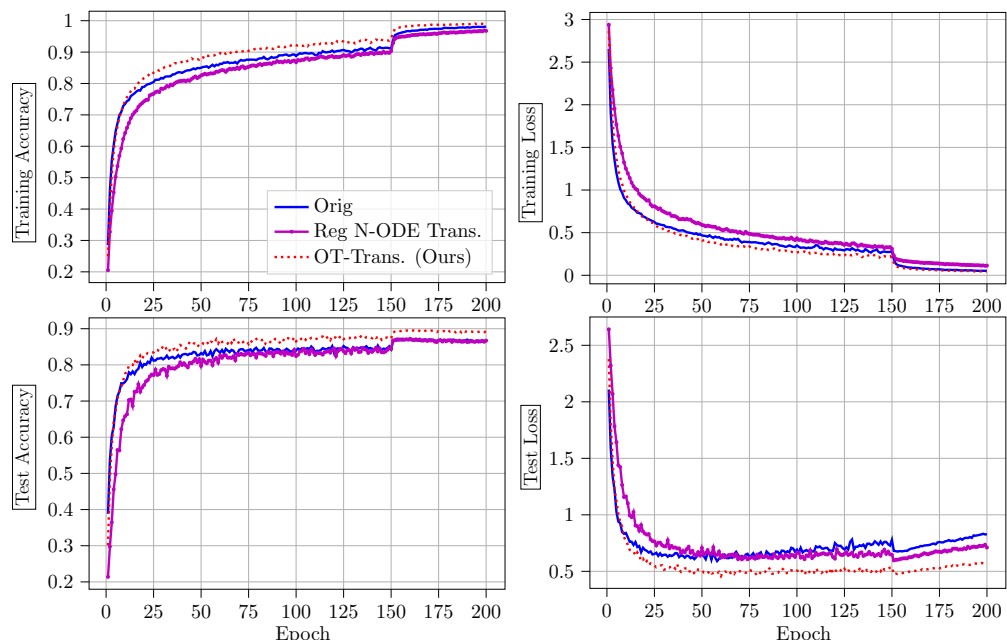

Figure 2: Accuracy and data-fitting loss for the point cloud experiment (averaged over five trials)

**MNIST Classification.** Following [74], the baseline model ViT has one Transformer block with a single-head self-attention layer and no fully-connected layer. Since it has only one Transformer block, N-ODE Transformer and our OT-Transformer share the same formulation, and we report the results as OT-Transformer.

The OT-Transformer uses the same model architecture as the baseline model, except that the hidden dimensions $d$ and $k$ of the self-attention layer are reduced to 64 from 128. This reduces the number of parameters by over $80\%$. Adapting from the baseline setting, the patch size is $7 \times 7$. We use an Adam optimizer. The number of epochs is 45 and the batch size is 100. The learning rate is set to $5 \times 10^{-4}$ for the first 35 epochs, then decreased to $5 \times 10^{-5}$ until the 41st epoch, at which point it is reduced to $5 \times 10^{-6}$. For OT-Transformer, the regularization hyperparameter is $\lambda = 0.01$ as it provides the optimal performance in our tests. We use $T = 1$ and a total of 20 time steps for the numerical integration.

The experiments are conducted over five random trials. The best test accuracies are reported in Table 7. OT-Transformer demonstrates significant improvements over the baseline in both accuracy and model efficiency. The baseline model achieved a test accuracy of 93.0%. The unregularized OT-Transformer improves the test accuracy to 96.8%, although it uses a much smaller model architecture. The transport cost regularization further improves the test accuracy to 97.1% while maintaining the same reduced parameter count. Notably, OT-Transformer also exhibits significantly lower standard deviation across five trials when compared to the baseline and unregularized model, indicating enhanced stability and reliability in its performance. Interestingly, when we compare the learning curves of the unregularized and regularized OT-Transformers in Figure 3, we observe that including the transport cost regularization also reduces the training loss for data-fitting and accuracy.

Table 7: Number of parameters for the Transformer blocks, best and final test accuracies (with standard deviation) across five trials for the MNIST image classification experiment.

| Method/Exp. | Para. Count | Best Test Accuracy | Final Test Accuracy |
|---|---|---|---|
| Baseline | 93k | $93.0\% \pm 0.69\%$ | $93.0\% \pm 0.67\%$ |
| Unreg. OT-Trans. | 18k | $96.8\% \pm 0.23\%$ | $96.8\% \pm 0.25\%$ |
| OT-Trans. (Ours) | 18k | $\mathbf{97.1\% \pm 0.16}\%$ | $\mathbf{97.1\% \pm 0.15}\%$ |

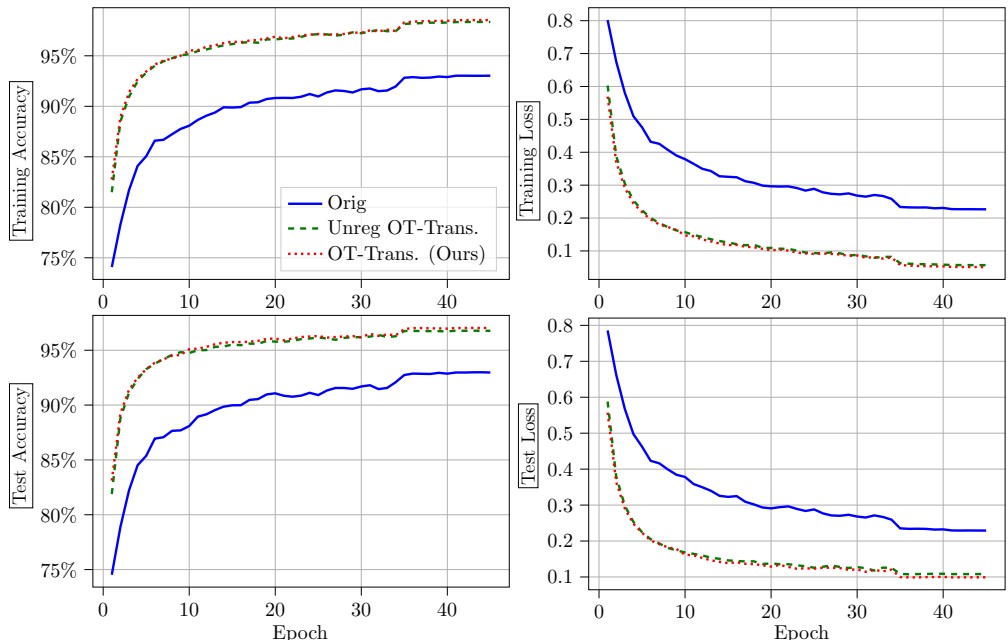

Figure 3: Accuracy and data-fitting loss for the MNIST image classification experiment (averaged over five trials)

**Cats and Dogs Classification.** We again use ViT. The patch size is $16 \times 16$. We use an Adam optimizer. The learning rate is $3 \times 10^{-5}$. The number of epochs is 250, and the batch size is 64. The hidden dimensions $d$ and $k$ are 128. For the baseline model, it has 6 Transformer blocks. For the continuous-time models, the number of Transformer blocks is reduced to 5; this reduces the number of parameters for the Transformer blocks by around 20%. For the regularized N-ODE Transformer and OT-Transformer, the regularization hyperparameters are $\lambda = 0.005$ and $\lambda = 0.01$, respectively, as they provide the optimal performance in our tests. We use $T = 1$ and a total of 20 time steps for the numerical integration.

We report in Table 8 the test accuracies over different seeded trials. We observe again that our OT-Transformer has the best performance and obtains a test accuracy of $79.0\%$, improving from the baseline's $77.6\%$. The standard deviation of the test accuracy, at $0.31\%$, is significantly lower than the baseline value of $0.86\%$, showing our proposed approach is more robust and reliable. We also observe that incorporating the transport cost regularization improves generalization and stability of OT-Transformer; without it, the average and standard deviation of test accuracy worsen to $78.2\%$ and $0.39\%$, respectively. Both the unregularized and regularized N-ODE Transformers report a test accuracy of $75.6\%$, which is worse than the baseline model, making them undesirable methods for the problem. Unlike our model, incorporating the regularization also has little effect on the performance of N-ODE Transformer. This is likely due to the incompatibility of N-ODE Transformer and the regularizatio. We report the learning curves in Figure 4. When we compare the learning curves of the unregularized and regularized OT-Transformers, we see that including the transport cost regularization also improves the training loss for data-fitting and accuracy.

Table 8: Number of parameters for the Transformer blocks, best and final test accuracies (with standard deviation) across three trials for the cats and dogs image classification experiment.

| Method/Exp. | Para. Count | Best Test Accuracy | Final Test Accuracy |
|---|---|---|---|
| Baseline | 1.77M | $79.3\% \pm 0.52\%$ | $77.6\% \pm 0.86\%$ |
| Unreg. N-ODE Trans. | 1.48M | $76.4\% \pm 0.37\%$ | $75.6\% \pm 0.48\%$ |
| Reg. N-ODE Trans. | 1.48M | $76.4\% \pm 0.30\%$ | $75.6\% \pm 0.03\%$ |
| Unreg. OT-Trans. | 1.48M | $78.8\% \pm 0.63\%$ | $78.2\% \pm 0.39\%$ |
| OT-Trans. (Ours) | 1.48M | $\mathbf{79.5\% \pm 0.46\%}$ | $\mathbf{79.0\% \pm 0.31\%}$ |

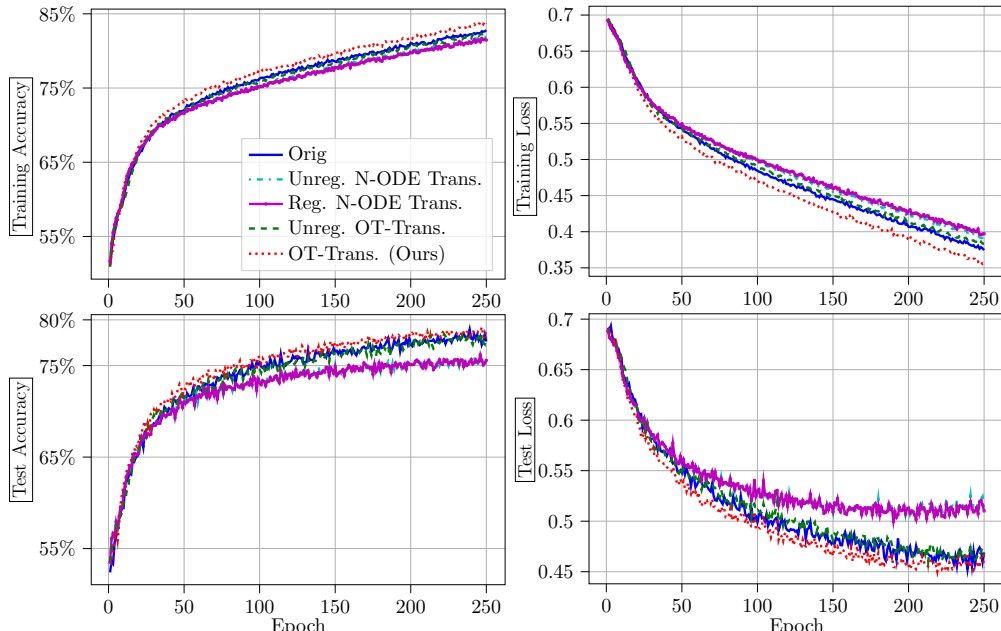

Figure 4: Accuracy and data-fitting loss for the cats and dogs image classification experiment

**Sentiment Analysis.** We use an identical baseline Transformer architecture as in [74], which has six layers of Transformer blocks. The OT-Transformer counterpart has only 3 layers, reducing the number of parameters of the Transformer blocks by half. We use an Adam optimizer with 15 epochs. The learning rate is $1 \times 10^{-4}$ for the first 12 epochs and $1 \times 10^{-5}$ afterward. The batch size is 64. The hidden dimensions $d$ and $k$ are 256. The batch size is 64. For both the regularized N-ODE Transformer and OT-Transformer, the regularization hyperparameter is $\lambda = 0.5$, as it provides the optimal performance in our tests. We use $T = 1$ and a total of 8 time steps for the numerical integration.

We repeat the experiment for five random trials. In all trials, the unregularized N-ODE Transformer and OT-Transformer experienced issues with exploding gradients, resulting in NaN outputs. In order to estimate how the unregularized model would perform under more stable conditions, we impose a slight transport cost with $\lambda = 0.001$. We note that the continuous-time models with slight and standard regularization completed all trials without issues. This shows the effectiveness of the transport cost regularization in stabilizing the training process and avoiding exploding gradients.

The best test accuracies are reported in Table 9. The baseline architecture achieved a test accuracy of 83.9%. The N-ODE Transformers with slight and standard regularization report a test accuracy of 83.6% and 83.9%, respectively, which are not better than the baseline model. The N-ODE Transformer with slight regularization reports a test accuracy of 83.6%. With a standard regularization, the test accuracy slightly increases to 83.9%. However, both results are not better than that of the baseline model. The OT-Transformer with slight regularization reported a test accuracy of 82.7%, which is

subpar compared to the baseline model. On the other hand, the standard OT-Transformer achieves the best test accuracy of 84.6%, which is 0.7% higher than the baseline model, in spite of using a smaller model. The test accuracy is also 0.7% higher than that of the N-ODE Transformer's. We note that with the incorporation of transport cost, the accuracy of N-ODE Transformer is improved by only 0.3%. In contrast, the accuracy of OT-Transformer is boosted by 1.9%. Again, this is likely due to that our continuous-time formulation is inherently more suited for transport cost regularization than that of N-ODE Transformer.

The learning curves are reported in Figure 5. When we compare the results of the unregularized and regularized OT-Transformers, we see that the regularization effectively reduces overfitting by increasing training loss while simultaneously lowering test loss. Overall, we see that the combination of our continuous-in-time formulation and transport cost regularization enhances parameter efficiency and generalization of Transformers.

Table 9: Number of parameters for the Transformer blocks, best and final test accuracies (with standard deviation) across five trials for for the sentiment analysis experiment. $^*$: The unregularized continuous-time models experienced gradient explosion. And we estimate their performance by using a slight regularization $\lambda = 0.001$.

| Method/Exp. | Para. Count | Best Test Accuracy | Final Test Accuracy |
|---|---|---|---|
| Baseline | 4.74M | $83.9\% \pm 0.26\%$ | $\mathbf{83.7\% \pm 0.21\%}$ |
| Unreg. N-ODE Trans. | 2.37M | $83.6\% \pm 0.40\%^*$ | $83.4\% \pm 0.40\%^*$ |
| Reg. N-ODE Trans. | 2.37M | $83.9\% \pm 0.48\%$ | $83.5\% \pm 0.84\%$ |
| Unreg. OT-Trans. | 2.37M | $82.7\% \pm 0.38\%^*$ | $82.1\% \pm 0.89\%^*$ |
| OT-Trans. (Ours) | 2.37M | $\mathbf{84.6\% \pm 0.55\%}$ | $83.7\% \pm 0.86\%$ |

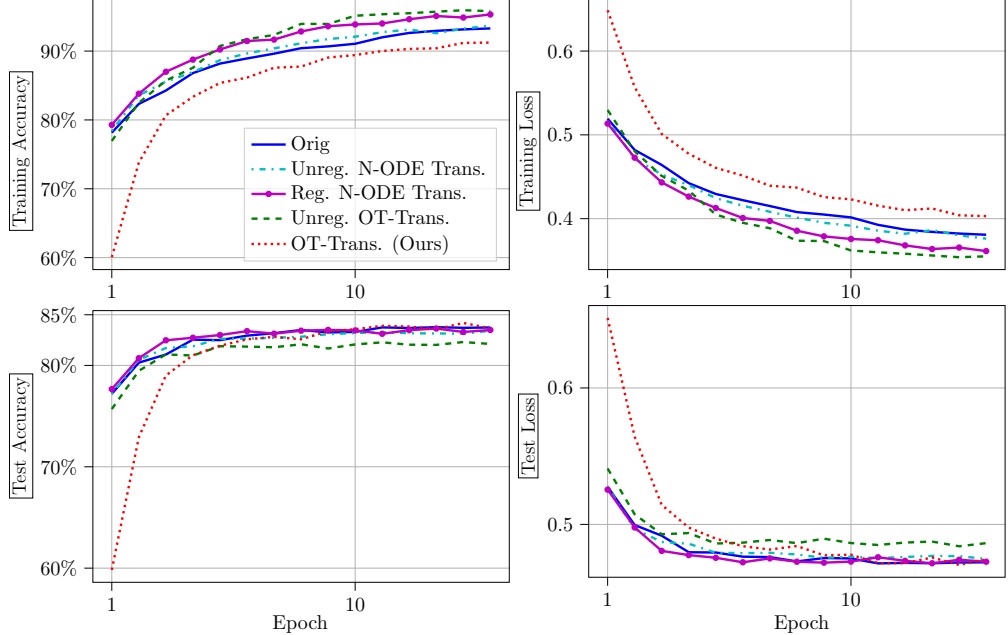

Figure 5: Accuracy and data-fitting loss for the sentiment analysis experiment

**Text Generation: nanoGPT on Shakespeare (Character-level) Dataset.** We use the nanoGPT implementation provided by [48] as the baseline model. The baseline architecture uses a Transformer model with 6 Transformer blocks, 6 self-attention heads and hidden dimension of 384, resulting in a total parameter count of 10.65 million. In contrast, for the OT-Transformer model, we reduce both the number of Transformer blocks and number of attention heads to 5 and hidden size to 320, in total this reduces the total parameter count to 6.16 million. We use 10 time steps for the numerical integration

and $\lambda = 1$. We run a total of $5000$ training iterations. The learning rate decays from $1 \times 10^{-3}$ to $1 \times 10^{-4}$ over the course of training. Model performances are measured using the test loss in all tests. The reported results are reported in Table 10 and averaged over 3 random trials. We highlight two main findings from the results. First, our proposed model displays significant improvement in parameter efficiency compared to the baseline solution, this is consistent with earlier results from different classification examples. Second, our model outperforms the baseline model in both the best recorded test loss and final step test loss, more notably our model ensures generalization while the baseline model overfits and shows a decline in performance, this coincides with our theoretical results and further highlights the importance of regularization.

Table 10: Number of parameters for the models, best and final test losses (with standard deviation) and additional LLM metrics across three trials for for the nanoGPT experiment. Rouge and Bleu scores are omitted as less meaningful for character-level experiments; see [20, 32].

| Method | Para. Count | Best Test Loss | Final Test Loss | Perplexity | BERT-P | BERT-R | BERT-F1 |
|--------|-------------|----------------|-----------------|------------|--------|--------|---------|
| Baseline | 10.65M | $1.47 \pm 0.005$ | $2.68 \pm 0.006$ | 17.26 | 74.4 % | 81.1 % | 77.6 % |
| Ours | 6.16M | $\mathbf{1.44 \pm 0.004}$ | $\mathbf{1.44 \pm 0.005}$ | **4.40** | **74.8%** | **81.2%** | **77.9%** |

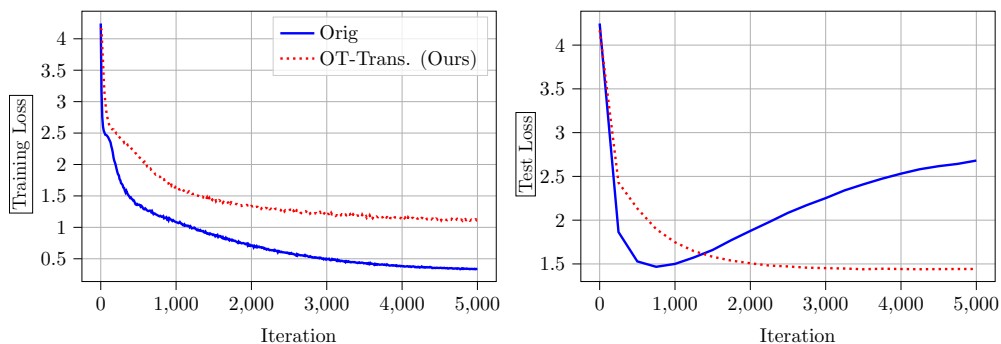

Figure 6: Training and test loss for the nanoGPT on Shakespeare (character-level) experiment.

**Interpretation of Model Uncertainty through the Perplexity Metric** We provide a more detailed explanation of model robustness in terms of the prediction uncertainty of a trained model. Note that the perplexity metric, derived from the test loss, measures the discrepancy between the predicted distribution and the underlying true distribution. In simple terms, a model with perplexity 10 behaves as if it is equally uncertain among 10 plausible choices, even if the actual distribution is not uniform. Thus, a lower perplexity is usually more desirable in practice. We derive Table 11 from Table 3 by including the additional perplexity metric. A visualization of the perplexity metric is also shown in Figure 7. It is evident that our model yields much smaller perplexity values across different noise levels, indicating a much lower level of uncertainty in the model's predictions.

Table 11: Test loss and perplexity metrics of the nanoGPT model under noise, where noise is introduced as random text replacement.

| Experiment | Metric/replace rate | 0.0 | 0.005 | 0.01 | 0.05 | 0.1 |
|------------|---------------------|-----|-------|------|------|-----|
| | Loss Base. | 2.68 | 2.78 | 2.88 | 3.65 | 4.60 |
| **NanoGPT** | Loss Ours | **1.44** | **1.49** | **1.55** | **1.95** | **2.42** |
| (text replace) | Perplexity Base. | 14.59 | 16.12 | 17.81 | 38.47 | 99.48 |
| | Perplexity Ours | **4.22** | **4.44** | **4.71** | **7.03** | **11.25** |

**Text Generation: GPT-2 on Shakespeare (Word-level) Dataset.** We next perform experiments using the more popular GPT-2 architecture on the Shakespeare (Word-level) dataset. We use the same implementation as in the first experiment, which is based on [48]. The goal for the experiment is to

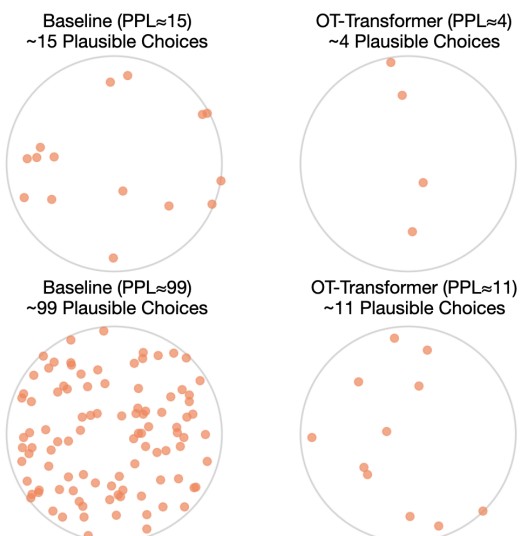

Figure 7: Visualization of the perplexity metric for baseline and our OT-Transformer

test the generalizability of our proposed approach to large models of over 100M total parameters. The baseline architecture has in total 12 layers, 12 attention heads and embedding size of 768, the models has 123.7M parameters in total. For above mentioned reasons we use the same exact architecture for OT-Transformer testing. Its also worth mentioning that the GPT-2 model is in fact overparametrized for the specific task, which also means reducing the model size is less significant for the task and should not be the focus. We use 10 time steps for the numerical integration and set $\lambda = 1$ for the example. We run a total of 500 iterations for training. The learning rate decays from $6 \times 10^{-4}$ to $6 \times 10^{-5}$ over the course of training. Model performances are measured using the test loss in all tests. The reported results are reported in Table 12 and averaged over 3 random trials.

The results indicate that our proposed approach comes out ahead in both the best documented test loss as well as the test loss in the last iteration. This shows the efficacy of our method extends to large-scale Transformer models. We also point out that while the baseline model sees fluctuations in performance as training progresses, with proper regularization our method is more stable as training progresses.

Table 12: Number of parameters for the models, best and final test losses (with standard deviation) across three trials for the GPT-2 and Shakespeare dataset experiment. We use test loss as the measure of model performance.

| Method/Exp. | Para. Count | Best Test Loss | Final Test Loss |
|---|---|---|---|
| Baseline | 123.7M | $4.91 \pm 0.001$ | $5.18 \pm 0.032$ |
| OT-Trans. (Ours) | 123.7M | $\mathbf{4.87 \pm 0.035}$ | $\mathbf{4.96 \pm 0.012}$ |

**Text Generation: GPT-2 on OpenWebText Dataset.** Lastly, We conduct a large-scale experiment on text generation using GPT-2, trained on the OpenWebText dataset [30]. The dataset is approximately 17GB in size, with the training data containing around 9 billion tokens and the test data containing about 4 million tokens. We use the GPT-2 model as the baseline, the Transformer architecture has in total 12 layers, 12 attention heads and embedding size of 768, as a result, the model has 123.7M parameters in total. We use the same architecture to parametrize OT-Transformer. For OT-Transformer, we use 10 time steps for the numerical integration and set $\lambda = 1$. We run a total of $60,00$ iterations for training. The learning rate decays from $6 \times 10^{-4}$ to $6 \times 10^{-5}$ over the course of training. Given the large scale of the experiment, we perform only one seeded trial for each model. We use the example to test the performance of our method on not only large models but also large datasets.

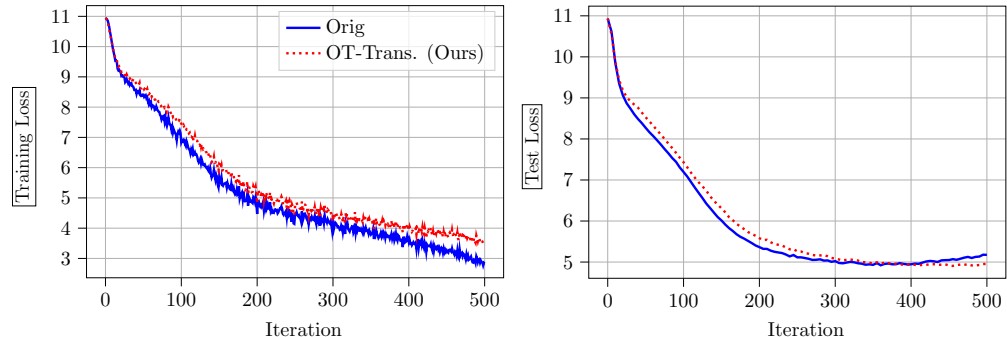

Figure 8: Training and test loss for the GPT-2 on Shakespeare (word-level) experiment.

We present the results in Table 13, we see that our OT-Transformer outperforms the baseline GPT-2 by a sizable margin in both best test loss and final test loss. These results highlight the ability of our model to outperform established baselines in large-scale and realistic experimental settings.

Table 13: Number of parameters for the models, best and final test accuracies and LLM metrics for the GPT-2 and OpenWebText dataset (9 billion tokens) experiment. Both the baseline and our OT-Transformer use a model with 123.7 million parameters.

| Model | B. Loss | F. Loss | Perplexity | BLEU | ROUGE-1 | ROUGE-2 | ROUGE-L | BERT-P | BERT-R | BERT-F1 |
|---|---|---|---|---|---|---|---|---|---|---|
| Baseline | 3.20 | 3.21 | 26.09 | 12.22% | 57.30% | 17.07% | 35.17% | 80.77% | 83.31% | 82.02% |
| Ours | **2.90** | **2.91** | **19.56** | **14.07**% | **59.46**% | **18.99**% | **38.05**% | **81.91**% | **84.27**% | **83.07**% |

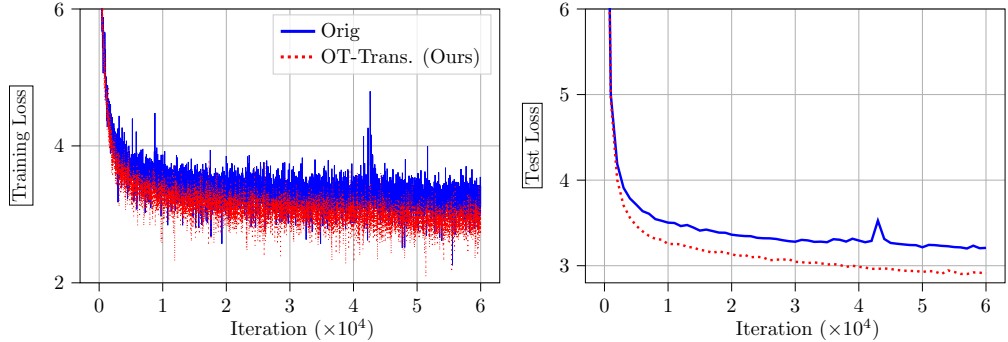

Figure 9: Training and test loss for the GPT-2 on OpenWebTest experiment. The Test loss is computed every 1,000 iterations.

