# OpenReview forum: "Optimal Control for Transformer Architectures: Enhancing Generalization, Robustness and Efficiency"
_NeurIPS.cc/2025/Conference — NeurIPS 2025 poster_

### Official Review · Reviewer_PviE · 2025-07-01

**Clarity:** 3
**Significance:** 3
**Originality:** 4
**Rating:** 5
**Confidence:** 4

**Summary:**

This paper investigates the optimal control perspective of Transformer architectures. Building on this perspective, the authors introduces an OT-regularizer which seems to help the performance of the Transformer training in practice.

**Questions:**

See above.

**Ethical Concerns:**

["NO or VERY MINOR ethics concerns only"]

**Final Justification:**

The authors clarified the relevance of the paper. I'd like to maintain my positive assessment.

**Limitations:**

See above.

**Paper Formatting Concerns:**

None to my knowledge

**Quality:**

3

**Strengths And Weaknesses:**

The perspective presented in this paper is nice and fresh. Also, I appreciate the OT-based regularizer working nicely in practice.

A few suggestions/questions:

1. First, I would appreciate as a reader if the authors really highlight the main difference resulted by the OC-based perspective. I get it from reading the paper, but it would really clear up the main contributions if the authors really walk through an example (e.g. GPT training) how the OC-based perspective together with OT-regularize leads to a different training framework.

2. Also, I would like to see the validation loss curve for OT-transformer and compare it with standard Transformer to see how the training dynamics differ.

3. Along the similar vein as 1), please show how the OT-based regularizer is implemented in practical Transformer training. You should clearly present the equation of the regularizer, or pseudocode or some sort.

4. Please share the full training details in the appendix. (Optimizer hyperparameters etc)

5. I noticed that for some of the experiments you chose smaller models for OT-Transformer, but for GPT2 training, you used the same size transformers. Is that any particular reason for this? I wonder what happens if you use smaller models for GPT2 training as well?

---

> ### Author Rebuttal · Authors · 2025-07-30
>
> We sincerely thank the reviewer for their time, effort, and constructive feedback. They have helped us improve and strengthen our paper. We also thank them for acknowledging the novelty of our work. We address the reviewer's concerns in bullet points as follows. We also conducted additional experiments to address the comments of the reviewers.
> ___
> **Concerns 1 and 2: How OT regularizer lead to a different training framework? Show how the model is implemented, by equation or pseudocode**
>
> **Responses 1 and 2:** Figure 1 of our paper gives an overview of our approach, and how it can be adapted from existing transformer architecture. To further clarify this, we here provide a pseudocode in algorithm format that details how our framework can be easily implemented.
>
> Since uploading images and links are prohibited, we have made our best effort to present the pseudocode here. We will include the pseudocode (in proper LaTeX algorithm formatting, with the standard Transformer and our model compared side-by-side) to the camera-ready version of the paper.
>
> **Pseudocode: Forward Propagation of Standard Transformer vs OT-Transformer**
>
> Inputs: Input $\mathbf{X}_0$, Transformer model $f = f_D \circ \dots \circ f_1$, terminal time $T$, regularization parameter $\lambda$, number of numerical integration steps $M$
>
> Define: Step size $\Delta t = \frac{T}{M}$
>
> **Standard Transformer**
>
> For $i = 1, 2, \dots, D$:
>
>   $\mathbf{X}{i} \leftarrow \mathbf{X}{i-1} + f_i(\mathbf{X}_{i-1})$
>
> Return: output $\mathbf{X}_D$
>
> **OT-Transformer**
>
> Initialize: $\mathbf{X}(0) = \mathbf{X}_0$, $\text{reg} = 0$
>
> For $t = \Delta t, 2 \Delta t, \dots, T$:
>
>   Compute $f(\mathbf{X}(t))$ *Adapt from standard Transformer*
>
>   $\mathbf{X}(t + \Delta t) \leftarrow \mathbf{X}(t) + \Delta t \cdot f(\mathbf{X}(t))$
>
>   $\text{reg} \leftarrow \text{reg} + \Delta t \cdot \|\|f(\mathbf{X}(t)) \|\|_F^2$
>
> Return: output $\mathbf{X}(t)$, regularization $\lambda \cdot \text{reg}$
> ___
> **Concern 3: Show validation loss curve**
>
> **Response 3:** Validation curves for all tested examples are reported in Appendix G, specifically, see Figures 2-8.
> ___
> **Concern 4: Share full training details**
>
> **Response 4:** The full training details for all examples are reported in Appendix G. The reader should be able to reproduce our results using the information shared. We will also release our source code on Github upon publication of this work to further ensure reproducibility.
> ___
> **Concern 5: Why GPT-2 uses a model of the same size**
>
> **Response 5:** Overall, our model consistently outperforms the baseline models, whether it uses the same or a reduced model size.
>
> Since the GPT-2 model is of larger size (>100M parameters), we use it to demonstrate that our approach can also generalize to larger models and datasets.
>
> For the other models, we reduce the number of parameters to demonstrate that our model is more parameter-efficient. This approach follows common practice in the continuous-time modeling literature; see [1-3] for example.
>
> Additionally, we conducted a new NanoGPT experiment in which OT-Transformer uses the same model size. The overall results, reported below, clearly show that our approach achieves better performance with both full-sized and reduced-sized models.
>
> **NanoGPT experiment**
> | Model    | Model size | Test loss        |
> | -------- | ---------- | ---------------- |
> | Baseline | 10.65M     | 2.68 ± 0.006     |
> | Ours     | 10.65M     | **1.44 ± 0.007** |
> | Ours     | 6.16M      | **1.44 ± 0.005** |
> ___
> [1] Chen, R. T., Rubanova, Y., Bettencourt, J., & Duvenaud, D. K. (2018). Neural ordinary differential equations. Advances in neural information processing systems, 31.
>
> [2] Grathwohl, W., Chen, R. T., Bettencourt, J., Sutskever, I., & Duvenaud, D. (2019). FFJORD: Free-Form Continuous Dynamics for Scalable Reversible Generative Models. In International Conference on Learning Representations.
>
> [3]  Massaroli, S., Poli, M., Park, J., Yamashita, A., & Asama, H. (2020). Dissecting neural odes. Advances in neural information processing systems, 33, 3952-3963.

---

> > ### Comment · Reviewer_PviE · 2025-08-02
> > **Thank you**
> >
> > Thank you for your response.
> > I did see from the GPT training curves that the benefit mostly comes from preventing overfitting instead of straight-up faster convergence. That said, the presented method might not be too helpful for large-scale pretraining (where val loss is train loss). That said, I think the empirical benefits seem validated across various domain, I will maintain my score.

---

> ### Author Response · Authors · 2025-08-02
> **Author Response**
>
> Thank you for your response. **Our approach improves performance through means beyond just reducing overfitting. Out of all seven experiments, there is only one experiment (NanoGPT) with serious overfitting issues with the baseline, and our model improvement is from reduced overfitting. However, this experiment is of small-scale, and is not used in realistic large-scale pre-training.**
>
> We would like to refer you to Figure 8, which shows the learning curves for the GPT2 on OpenWebText experiment. The model has >100M parameters and the experiment is larger-scale and **best reflects a realistic pre-training setting. The figure shows clearly that our model achieves better training and test loss throughout the training iterations.**
>
> The same pattern (lower training and test losses throughout the training process) also holds for our three image classification and point cloud classification experiments; see Figures 2, 3, and 4.
>
> We also re-iterate that the improved generalization is universal and supported by our theory; see Theorems 1-3. The complete derivations are given in Appendices B-F.  **Our theory is a key contribution. It provides novel theoretical tools to analyze Transformers and sets us apart from existing work.** We also provided additional experiments to validate our theory; see Response 1 to Reviewer ZUmk.

---

### Official Review · Reviewer_SwmS · 2025-07-03

**Clarity:** 2
**Significance:** 2
**Originality:** 3
**Rating:** 4
**Confidence:** 3

**Summary:**

In this paper, the authors bring insights from optimal control theory to Transformer model training. With a plug-and-play modification, Transformers are formulated in a continuous-time setting and incorporate regularization during training. Seven experiments covering point-cloud, image, sentiment, and text generation tasks show that OT-Transformer either matches or exceeds the baseline while often using fewer parameters.

**Questions:**

1. Adding evaluations across both smaller and larger model sizes, with and without OT-Transformer, may help assess the effectiveness of the proposed method.

2. Fixing the hyperparameters and using fixed random seeds would improve reproducibility and reliability.

3. Providing a guideline on how to reduce the model size when applying OT-Transformer would be helpful.

**Ethical Concerns:**

["NO or VERY MINOR ethics concerns only"]

**Final Justification:**

During the rebuttal period, the authors have answered most of my questions. These include:

1. The authors have provided pseudo code that helps clarify the implementation of the proposed algorithm.
2. The authors have pointed out that the selection of hyperparameters is based on previous works.
3. A series of experiments have been conducted to support the claims on robustness and out-of-distribution generalization.
4. The authors have reported the training time compared to both the original model and the Standard Continuous-Time Transformer, providing sufficient information to evaluate the pros and cons.
5. Although I am still not convinced by the discussion on model size, I believe it does not significantly affect the contribution of this work.

For the above reasons, I am willing to raise my score to borderline accept.

**Limitations:**

See Weaknesses.

**Quality:**

3

**Strengths And Weaknesses:**

1. The authors claim that the proposed method is better at avoiding overfitting. However, it is difficult to determine whether the evaluation results support this claim, as the model size is reduced. The improved performance or reduced overfitting may be due to the smaller number of parameters rather than the new training algorithm.

2. The authors also claim that stable forward propagation results in robustness to input perturbations and out-of-distribution generalization. However, there is no evaluation provided to support this claim.

3. There is no pseudocode, which makes it difficult to understand how the formulation is implemented and whether it is correct.

4. There is no discussion or ablation study on the selection of hyperparameters such as lambda, T, and the total number of time steps. The choice of hyperparameters appears cherry-picked, as the number of training epochs and the learning rate vary significantly across datasets. Additionally, there is no general rule provided for how or why the model size should be reduced for OT-Transformer: in the MNIST experiment, the model size is reduced by 5x, while in the Shakespeare dataset, there is no reduction.

---

> ### Author Rebuttal · Authors · 2025-07-30
>
> We sincerely thank the reviewer for their time, effort, and constructive feedback. They have helped us improve and strengthen our paper. We address the reviewer's concerns in bullet points as follows. We also conducted additional experiments to address the comments of the reviewers.
> ___
> **Concern 1: Difficult to determine whether the evaluation results support that our model is better at avoiding overfitting, since the model size is reduced. The improved performance/reduced overfitting may be due to reduced number of parameters**
>
> **Response 1: Our seven diverse experimental results (in Appendix G) suggest the opposite. Our model does not suffer from overfitting issues. Our model is more parameter-efficient. We explain through the following details.**
>
> 1. In our experiments, our model **achieves both lower training and test losses (on unseen data) consistently across different examples** (see Appendix G), despite having fewer parameters. **These results show that the performance improvement is due to parameter efficiency, not overfitting.**
>
> 2. We point the reviewer to the two GPT2 experiments presented, in which our model and the baseline have the same number of parameters, yet our model consistently performs better. This directly supports that our model is better at avoiding overfitting.
>
> 3. We remark that reducing model size to demonstrate parameter efficiency is a common practice in the continuous-time modeling literature—for example, continuous normalizing flows [3-5].
>
> 4. To further show that we do not reduce model size to avoid overfitting, we conducted a new NanoGPT experiment, where OT-Transformer uses the same model size as the baseline, and we report the overall results below. Under repeated trials, **our models outperforms the baseline, whether the model size is matched or reduced.**
>
> **NanoGPT experiment**
> | Model    | Model size | Test loss        |
> | -------- | ---------- | ---------------- |
> | Baseline | 10.65M     | 2.68 ± 0.006     |
> | Ours     | 10.65M     | **1.44 ± 0.007** |
> | Ours     | 6.16M      | **1.44 ± 0.005** |
>
> ___
> **Concern 2: The authors also claim that stable forward propagation results in robustness to input perturbations and out-of-distribution generalization**
>
> **Response 2: These are not merely claims; they are rigorously and mathematically proven results.** See Theorems 1-3 and Appendix F. We prove that our training formulation leads to stable forward propagation, which in turn leads to improved robustness and generalization. Specifically, we derive deterministic generalization guarantees when the training or test distributions are perturbed. Moreover, we derive a non-asymptotic and statistical generalization guarantee for out-of-distribution test datasets. Importantly, these guarantees can be controlled through the strength of the OT regularization.
>
> On the other hand, these bounds do not hold in general for a standard Transformer setup without any type of optimal transport regularization; see Theorems 1 and 2 in Section 4. Because the training is ill-posed and can lead to models with highly irregular (e.g. exploding) hidden states. This can also explain why we have a more stable training loss curve in the GPT2 WebText experiment in Figure 8, where the model is deep and more prone to training instability. Because **our training formulation always leads to a unique model with highly regular hidden states**, this can prevent the hidden states and network weights from exploding during training and resulting in NaNs; see the derivations in Appendices D3-D6.
>
> Also, our extensive results across seven experiments show that our model has improved robustness and generalization; see Tables 1 and 2 and Appendix G where our approach consistently leads to improved performance, lower variance and better training stability.
> ___
> **Concern 3: No evaluation provided to support that stable forward propagation results in robustness and out-of-distribution generalization**
>
> **Response 3:** We appreciate this useful feedback. **This concern is addressed by our four additional experiments, where our model consistently demonstrates superior robustness and out-of-distribution generalization over the baseline in all settings.** Due to space constraints, please refer to our response 1 to Reviewer ZUmk.
> ___
> **Concern 4: No psuedocode to explain**
>
> **Response 4:** We thank the reviewer for the helpful suggestion, we have added a pseudocode that helps clarify and explain our proposed approach. Due to space constraints, please refer to our responses 1 and 2 to Reviewer PviE.
> ___
> **Concern 5: Discussion/ablation study on selecting hyperparameters**
>
> **Response 5:** Additional ablation study is performed to address the concern. Due to space constraints, please refer to our response 5 of Reviewer fc21.
> ___
> **Concern 6: Hyperparameters appear to be cherry-picked**
>
> **Response 6: We did not cherry-pick any hyperparameters.** The hyperparameters used in the Transformer models and experiments—such as the number of epochs, step size, batch size—follow the original settings from [1] and [2]. While these hyperparameters may appear different for different examples, they are determined by the original authors and not us.
>
> We also want to highlight that even though the hyperparameters used are not tailored for our approach, we consistently obtained superior results across seven diverse experiments, demonstrating the strength of our framework.
> ___
> **Concern 7: Use fixed random seeds**
>
> **Response 7:** As discussed in our paper, we use fixed random seed = 1,2,3,4,5 for our experiments to ensure reproducibility. Moreover, the reported results are averaged over multiple trials with different seeds to reduce variance and ensure our claim is accurate.
>
> The full training details for all examples are reported in Appendix G. The reader should be able to reproduce our results using the information shared. We will also release our source code on Github upon publication of this work to further ensure reproducibility.
> ___
> **Concern 8: Provide guidelines on how to reduce the model size**
>
> **Response 8:** We reduce the number of parameters to demonstrate that our model is more parameter-efficient. This approach follows common practice in the continuous-time modeling literature; see [3-5] for example.
>
> It is not necessary to reduce the model size. Specifically, using the same model size as the baseline can also achieve improved performance; see our GPT2 experiments in Appendix G. We also perform additional experiments on NanoGPT, where our model consistently outperforms the baseline, whether it is using the same or reduced model size. For the results, please refer to our response 1 to you.
> ___
> [1] Sander, M. E., Ablin, P., Blondel, M., & Peyré, G. (2022, May). Sinkformers: Transformers with doubly stochastic attention. In International Conference on Artificial Intelligence and Statistics (pp. 3515-3530). PMLR.
>
> [2] Karpathy, A. (2023). NanoGPT [Source code]. GitHub.
>
> [3] Chen, R. T., Rubanova, Y., Bettencourt, J., & Duvenaud, D. K. (2018). Neural ordinary differential equations. Advances in neural information processing systems, 31.
>
> [4] Grathwohl, W., Chen, R. T., Bettencourt, J., Sutskever, I., & Duvenaud, D. (2019). FFJORD: Free-Form Continuous Dynamics for Scalable Reversible Generative Models. In International Conference on Learning Representations.
>
> [5]  Massaroli, S., Poli, M., Park, J., Yamashita, A., & Asama, H. (2020). Dissecting neural odes. Advances in neural information processing systems, 33, 3952-3963.

---

> > ### Comment · Reviewer_SwmS · 2025-08-06
> >
> > Thank you for the detailed responses. I have some follow-up questions and comments:
> >
> > 1. Previously I didn’t know where the hyperparameters came from. After the authors’ clarification, I understand that the hyperparameters are from previous works, and I have checked the literature—the numbers are aligned.
> >
> > 2. The results for robustness and out-of-distribution generalization look promising.
> >
> > 3. I still don’t understand why, in some cases, you reduce the number of parameters by 5x, while in others there is less or no reduction. If it doesn't matter, why not just fix the reduction ratio?
> >
> > 4. Regarding your pseudocode, does it mean the computational cost is M times higher than the standard Transformer, since you compute f(X(t)) M times?

---

> ### Author Response · Authors · 2025-08-07
> **Response to Reviewer**
>
> Thank you for your response. We are glad that the concerns on hyperparameters have been addressed, and you find the experiments on robustness convincing.
>
> ___
>
> **Concern 9: I still don’t understand why, in some cases, you reduce the number of parameters by 5x, while in others there is less or no reduction. If it doesn't matter, why not just fix the reduction ratio?**
>
> **Response 9:** Parameter efficiency is one of the key advantages of our model.  We reduce the number of parameters to demonstrate that while still outperforming the baseline. The extent of reduction varies across experiments due to differences in the nature and complexity of the tasks. In some settings, we can outperform the baseline with a 5x parameter reduction, in other cases, a smaller reduction (e.g., 2x) is necessary to maintain competitiveness.
>
> We keep the model size only for the GPT-2 experiments. Since the GPT-2 model is of larger size (>100M parameters), we use it to demonstrate that our approach can also generalize to larger models and datasets.
> ___
>
> **Concern 10: Regarding your pseudocode, does it mean the computational cost is M times higher than the standard Transformer, since you compute f(X(t)) M times?**
>
> **Response 10:** The computational cost is not $M$ times higher. Here, $M$ denotes the number of numerical integration steps in the Euler scheme. We compute the forward of a Transformer $M$ times, which is an inherent feature of continuous-time models; see [1-8] for seminal work in this line of research. But the computational cost can be mitigated by using a smaller model thanks to our model's parameter efficiency. This also improves memory efficiency during inference.
>
> Moreover, our novel OT regularization guarantees highly regular hidden states; see Appendices D2-D7. It allows us to further reduce computational cost: by using fewer integration steps $M$ without sacrificing integration accuracy. For instance, in our response 5 to reviewer fc21, we tested our model's performance when $M$ varies. From the results, we see that the model performance is very stable with respect to $M$. Even when $M$ is small, our model still outperforms the baseline.
>
> The continuous-time formulation allows us to analyze and improve Transformer training and architectures. While it might introduce additional computational cost, thanks to the novel OT regularization, this added cost is much less than other continuous-time models [1-8]. More importantly, it enables a principled analysis (detailed in Appendices C-F) that offers significant theoretical insights and theory-grounded improvements, justifying the trade-off.
>
> ___
>
> [1] Chen, R. T., Rubanova, Y., Bettencourt, J., & Duvenaud, D. K. (2018). Neural ordinary differential equations. Advances in Neural Information Processing Systems, 31.
>
> [2] Grathwohl, W., Chen, R. T., Bettencourt, J., Sutskever, I., & Duvenaud, D. (2019). FFJORD: Free-Form Continuous Dynamics for Scalable Reversible Generative Models. In International Conference on Learning Representations.
>
> [3] Massaroli, S., Poli, M., Park, J., Yamashita, A., & Asama, H. (2020). Dissecting neural odes. Advances in Neural Information Processing Systems, 33, 3952-3963.
>
> [4] Dupont, E., Doucet, A., & Teh, Y. W. (2019). Augmented neural odes. Advances in Neural Information Processing Systems, 32.
>
> [5] Xia, H., Suliafu, V., Ji, H., Nguyen, T., Bertozzi, A., Osher, S., & Wang, B. (2021). Heavy ball neural ordinary differential equations. Advances in Neural Information Processing Systems, 34, 18646-18659.
>
> [6] Norcliffe, A., Bodnar, C., Day, B., Simidjievski, N., & Liò, P. (2020). On second order behaviour in augmented neural odes. Advances in Neural Information Processing Systems, 33, 5911-5921.
>
> [7] Rodriguez, I. D. J., Ames, A., & Yue, Y. (2022). Lyanet: A lyapunov framework for training neural odes. In International Conference on Machine Learning (pp. 18687-18703).
>
> [8] Zheng, S., Gao, Z., Sun, F. K., Boning, D., Yu, B., & Wong, M. D. (2024). Improving neural ode training with temporal adaptive batch normalization. Advances in Neural Information Processing Systems, 37, 95875-95895.

---

> > ### Comment · Reviewer_SwmS · 2025-08-08
> >
> > Thank you again for your response. Would it be possible for you to provide some quantitative values?
> >
> > > The computational cost is not M times higher.
> >
> > Could you clarify how much higher it actually is?
> >
> > > This added cost is much less than other continuous-time models.
> >
> > Could you specify how much less it is compared to those models?

---

> ### Author Response · Authors · 2025-08-09
> **Response to Reviewer**
>
> Thank you for your follow-up.
>
> In the following, we compare the average training time per iteration for the NanoGPT experiment.
>
> | Model                              | Time  |
> |------------------------------------|--------------------|
> | Baseline                           | 64.6ms               |
> | OT-Transformer                     | 175.0ms               |
> | Standard Continuous-time Transformer | 344.4ms               |
>
> The additional runtime is caused by the numerical integration. This shows a tradeoff between runtime and the benefits of parameter efficiency, improved test performance, and theoretically guaranteed robustness and generalization.
>
> The OT regularization induces regular hidden state dynamics. This allows fewer integration steps and halves runtime, while still outperforming the standard continuous-time Transformer; see the sensitivity test in our rebuttal and Appendix F.
>
> Additional training time (per epoch) is reported as follows. In the table below, we use the same number of integration time steps $M$ as the standard continuous-time Transformer. But our runtime can be improved by reducing $M$, thanks to the OT regularization. Due to time constraints, experiments with reduced $M$ are not conducted.
>
> | Method          | MNIST  | CatDog  | Sent. Analysis | Point Cloud |
> |-----------------|--------|---------|----------------|-------------|
> | Baseline        | 6.8s   | 76.9s   | 104.9s         | 21.8s       |
> | OT-Transformer  | 18.8s  | 204.8s  | 243.6s         | 60.0s       |

---

> > ### Comment · Reviewer_SwmS · 2025-08-09
> >
> > Thank you for the detailed information. I hope these can be included in the paper, as they will help readers understand the pros and cons and how much improvement your method can achieve compared to previous work. I have no further questions and will raise my score to borderline accept.

---

> > > ### Author Response · Authors · 2025-08-09
> > > **Thank you**
> > >
> > > Thank you very much for the continued discussion. Your valuable suggestions have greatly helped us improve the paper. We will incorporate all additional information and results from this rebuttal into the final draft.
> > >
> > > We sincerely appreciate your time, effort, and constructive feedback.

---

### Official Review · Reviewer_ZUmk · 2025-07-03

**Clarity:** 3
**Significance:** 3
**Originality:** 3
**Rating:** 3
**Confidence:** 3

**Summary:**

This paper formulates Transformer training as a continuous-time optimal control problem, leading to the proposal of OT-Transformer—a plug-and-play module that introduces an optimal transport (OT) based velocity regularization term. The authors provide theoretical results (well-posedness, stability bounds, distributional robustness) and validate the method across diverse domains including language, vision, and 3D point clouds.

**Questions:**

Are there any concrete empirical evidence supporting their theoretical claims, such as quantitative tests of stability or robustness under input perturbations? Additionally, how exactly is the continuous-time variable mapped to discrete Transformer layers—does each layer represent an equal time increment, or is the temporal analogy merely symbolic? It would also be helpful to clarify how the regularization strength λ is selected and whether the model's performance is sensitive to its tuning across different tasks or scales.

**Ethical Concerns:**

["NO or VERY MINOR ethics concerns only"]

**Final Justification:**

I still have some concerns, 1) It is similar to the point raised by Reviewer fc21. 'While the empirical evaluation is extensive, its plug-and-play design is still limited to the very simple tasks, like MNIST or binary classification, language generation is evaluated only based on the test loss that is usually not a completely fair point for benchmarking the language models. While the loss can be low, the generations themselves can be of different quality, for example, the model can loop into repetitions, and we are not able to notice it based on the pure loss.' 2) The robustness statement in the title and conclusion of the article is not rigorous. The experiment lacks verification on the corresponding dataset. 3）The paper needs some major revisions or additions after submission, which is also a factor in my final scoring.

**Limitations:**

Despite its theoretical appeal, the paper lacks direct empirical validation of its core theoretical claims—such as forward stability or distributional robustness—which remain largely untested in controlled experiments. Moreover, the continuous-time formulation and its interpretation in relation to discrete Transformer architectures are insufficiently clarified, weakening the credibility of the dynamical systems analogy.

**Quality:**

3

**Strengths And Weaknesses:**

Strengths：
This paper introduces a novel and theoretically grounded approach by casting Transformer training as a continuous-time optimal control problem, leading to a plug-and-play regularization scheme with provable stability and robustness properties. The proposed OT-Transformer is elegantly formulated, broadly applicable across modalities, and achieves consistent empirical improvements while reducing parameter counts, demonstrating both practical and theoretical value.

Weaknesses:
Despite its theoretical appeal, the paper lacks direct empirical validation of its core theoretical claims—such as forward stability or distributional robustness—which remain largely untested in controlled experiments. Moreover, the continuous-time formulation and its interpretation in relation to discrete Transformer architectures are insufficiently clarified, weakening the credibility of the dynamical systems analogy.

---

> ### Author Rebuttal · Authors · 2025-07-30
>
> We sincerely thank the reviewer for their time, effort, and constructive feedback. They have helped us improve and strengthen our paper. We are encouraged that the reviewer acknowledged the novelty, elegance, theoretical significance, and practical value of our work. We address the reviewer's concerns in bullet points as follows. We also conducted additional experiments to address the comments of the reviewers.
> ___
> **Concern 1: Lacks empirical validation for forward stability and distributional robustness**
>
> **Response 1:** We agree with the reviewer that additional experimentation can further support our theory, specifically Theorems 2 and 3 and Appendix F. We thank the reviewer for the helpful suggestion. We here conduct 4 additional experiments in which varying levels of random noise are injected into the test inputs. The noise was absent from the training data. This setup tests the model's **forward stability**, **distributional robustness**, and **out-of-distribution generalization**. **Our model consistently outperforms the baselines over all these tests across all noise levels.** Importantly, our model performs significantly better when the noise level is high. These additional results further support our theory and existing results.
>
> The following results with full implementation details will be added to the camera-ready version of the paper.
>
> **Experiment 1: Point could classification with point dropout** (commonly used noise; see [5-6]) (average test accuracy and std)
> | Drop rate | 0.0               | 0.01              | 0.05              | 0.1               | 0.2               | 0.5               |
> | --------- | ----------------- | ----------------- | ----------------- | ----------------- | ----------------- | ----------------- |
> | Baseline  | 86.6% ± 0.45%     | 86.6% ± 0.48%     | 85.8% ± 0.60%     | 84.3% ± 0.69%     | 76.9% ± 0.88%     | 34.5% ± 1.94%     |
> | Ours      | **89.3% ± 0.55%** | **89.3% ± 0.55%** | **88.8% ± 0.34%** | **87.6% ± 0.62%** | **83.9% ± 0.80%** | **55.4% ± 4.87%** |
>
> **Experiment 1: Point could classification with point dropout** (performance drop, lower is better)
> | Drop rate | 0.01     | 0.05     | 0.1      | 0.2      | 0.5       |
> | --------- | -------- | -------- | -------- | -------- | --------- |
> | Baseline  | **0.0%**     | 0.8%     | 2.3%     | 9.7%     | 52.1%     |
> | Ours      | **0.0%** | **0.5%** | **1.7%** | **5.4%** | **33.9%** |
>
> **Experiment 2: NanoGPT with random text replacement** (follows [7]) (average test loss and std)
> | Replace rate | 0.0              | 0.005            | 0.01             | 0.05             | 0.1              |
> | ------------ | ---------------- | ---------------- | ---------------- | ---------------- | ---------------- |
> | Baseline     | 2.68 ± 0.004     | 2.78 ± 0.004     | 2.88 ± 0.004     | 3.65 ± 0.004     | 4.60 ± 0.005     |
> | Ours         | **1.44 ± 0.005** | **1.49 ± 0.004** | **1.55 ± 0.003** | **1.95 ± 0.010** | **2.42 ± 0.022** |
>
> **Experiment 2: NanoGPT with random text replacement** (performance drop, lower is better)
> | Replace rate | 0.005    | 0.01     | 0.05     | 0.1      |
> | ------------ | -------- | -------- | -------- | -------- |
> | Baseline     | 0.10     | 0.20     | 0.97     | 1.92     |
> | Ours         | **0.05** | **0.11** | **0.51** | **0.98** |
>
> **Experiment 3: MNIST with Gaussian noise** (commonly used noise; see [8]) (average test accuracy and std)
> | Noise level | 0.0                | 0.01               | 0.05               | 0.1                | 0.2                | 0.5                |
> | ----------- | ------------------ | ------------------ | ------------------ | ------------------ | ------------------ | ------------------ |
> | Baseline    | 92.97% ± 0.67%     | 92.99% ± 0.66%     | 92.96% ± 0.69%     | 92.76% ± 0.72%     | 91.70% ± 0.68%     | 80.64% ± 1.58%     |
> | Ours        | **97.05% ± 0.15%** | **97.05% ± 0.16%** | **96.99% ± 0.18%** | **96.89% ± 0.15%** | **96.39% ± 0.11%** | **90.10% ± 1.10%** |
>
> **Experiment 3: MNIST with Gaussian noise** (performance drop, lower is better)
> | Noise level | 0.01      | 0.05      | 0.1       | 0.2       | 0.5       |
> | ----------- | --------- | --------- | --------- | --------- | --------- |
> | Baseline    | **-0.02%**    | **0.01%**     | 0.21%     | 1.27%     | 12.33%    |
> | Ours        | 0.00% | 0.06% | **0.16%** | **0.66%** | **6.95%** |
>
> **Experiment 4: MNIST with uniform noise** (commonly used noise; see [8]) (average test accuracy and std)
> | Noise level | 0.0                | 0.01               | 0.05               | 0.1                | 0.2                | 0.5                |
> | ----------- | ------------------ | ------------------ | ------------------ | ------------------ | ------------------ | ------------------ |
> | Baseline    | 92.97% ± 0.67%     | 92.99% ± 0.65%     | 92.98% ± 0.63%     | 92.90% ± 0.58%     | 92.64% ± 0.57%     | 90.02% ± 0.45%     |
> | Ours        | **97.05% ± 0.15%** | **97.03% ± 0.14%** | **97.00% ± 0.15%** | **96.97% ± 0.14%** | **96.79% ± 0.16%** | **95.57% ± 0.11%** |
>
> **Experiment 4: MNIST with uniform noise** (performance drop, lower is better)
> | Noise level | 0.01  | 0.05      | 0.1       | 0.2       | 0.5       |
> | ----------- | ----- | --------- | --------- | --------- | --------- |
> | Baseline    | **0.02%** | **-0.02%**    | **0.07%**     | 0.33%     | 2.95%     |
> | Ours        | **0.02%** | 0.05% | 0.08% | **0.26%** | **1.48%** |
> ___
> **Concern 2: The connection of continuous-time and discrete models are insufficiently clarified**
>
> **Response 2:** Transformer architectures include a skip-connection in each layer of their Transformer blocks. In particular, each layer can be written as
> $$ x_{t+1} = x_t + f(x_t; \theta_t). $$
> This can be seen as a forward Euler discretization of continuous dynamics. Specifically, we formulate the continuous dynamics as
> $$ \frac{d x(t)}{dt} = f(x(t), t; \theta), $$
> where $f$ is a Transformer model. This perspective is commonly adopted in continuous-time models and connects neural networks to dynamical systems. It enables the use of techniques from dynamical systems and has been central to the development of continuous-time models; see [1-4] for more details. **Due to space constraints, please refer to our responses 1 and 2 to Reviewer PviE for a pseudocode comparing the standard Transformer and our model.**
>
> As a part of our contribution, we show that directly applying a continuous-time formulation to Transformers results in an ill-posed training problem, which can be a major issue for large-scale training problems. The trained Transformers from this ill-posed training problem are sub-par in performance, which we demonstrated by both 1. empirical experiments (in Sections 1 & 5, and Appendix G) and 2. inspecting the regularity and optimality conditions of the training problem, which are a Hamilton-Jacobi-Bellman partial differential equation coupled with a continuity equation (the proofs are given in Appendices D1-D4).
>
> On the other hand, our novel theory demonstrates that the OT regularization leads to a highly regular trained model with generalization and robustness guarantees; see Section 4, and Appendices D5-D7 and F.
> ___
> **Concern 3: How does the continuous hidden states (of our model) relate to the discrete hidden states (of standard Transformer)**
>
> **Response 3:** The initial continuous hidden states ${\bf X}(0)$ is the same as the input of a standard Transformer. The terminal continuous hidden states ${\bf X}(T)$ is used as the output of our model. When a forward Euler discretization scheme with step size 1 is used, our model reduces to a standard Transformer. In this case, the continuous and discrete hidden states are the same. Thus, our model can be seen as a generalization of the standard Transformer architecture, with a continuous-depth architecture.
> ___
> **Concern 4: Clarify how the parameter $\lambda$ is tuned, whether it is sensitive**
>
> **Response 4:** Due to space constraints, please refer to our response 5 to Reviewer fc21.
> ___
>
> [1] Haber, E., & Ruthotto, L. (2017). Stable architectures for deep neural networks. Inverse problems, 34(1), 014004.
>
> [2] Chen, R. T., Rubanova, Y., Bettencourt, J., & Duvenaud, D. K. (2018). Neural ordinary differential equations. Advances in neural information processing systems, 31.
>
> [3] Grathwohl, W., Chen, R. T., Bettencourt, J., Sutskever, I., & Duvenaud, D. (2019) FFJORD: Free-Form Continuous Dynamics for Scalable Reversible Generative Models. In International Conference on Learning Representations.
>
> [4] Ruthotto, L. (2024). Differential equations for continuous-time deep learning. arXiv preprint arXiv:2401.03965.
>
> [5] Wang, Y., Sun, Y., Liu, Z., Sarma, S. E., Bronstein, M. M., & Solomon, J. M. (2019). Dynamic graph cnn for learning on point clouds. ACM Transactions on Graphics (tog), 38(5), 1-12.
>
> [6] Qi, C. R., Yi, L., Su, H., & Guibas, L. J. (2017). Pointnet++: Deep hierarchical feature learning on point sets in a metric space. Advances in neural information processing systems, 30.
>
> [7] Jin, D., Jin, Z., Zhou, J. T., & Szolovits, P. (2020, April). Is bert really robust? a strong baseline for natural language attack on text classification and entailment. In Proceedings of the AAAI conference on artificial intelligence (Vol. 34, No. 05, pp. 8018-8025).
>
> [8] Rafael C. Gonzalez and Richard E. Woods. 2006. Digital Image Processing (3rd Edition). Prentice-Hall, Inc., USA.

---

> ### Comment · Area_Chair_rUcM · 2025-08-08
>
> Dear Reviewer,
>
> With less than 24 hours remaining until the conclusion of discussion phase (August 8 11:59pm AoE), I'd like to again strongly encourage you to dedicate some time to review the authors' feedback and other reviews. Sharing your concluding thoughts with both the authors and the committee would be invaluable; however, given the limited time, please refrain from asking the authors to finish tasks that require a significant amount of time.
>
> Thank you for your continued engagement.
>
> Best regards,
>
> AC

---

### Official Review · Reviewer_fc21 · 2025-07-05

**Clarity:** 2
**Significance:** 3
**Originality:** 4
**Rating:** 4
**Confidence:** 3

**Summary:**

The paper introduces a novel framework that analyzes and augments Transformer models using optimal control theory and optimal transport (OT) regularization. The authors propose a novel regularization technique for training standard transformers as a continuous-time optimal control problem, where model depth corresponds to time and the loss is treated as a terminal cost. They train several versions of the so-called OT-Transformer architecture that adds regularization term to the training objective to design of the final layer. The method is plug-and-play, and does not require any complex modifications to the standard transformer architectures. The approach provides theoretical guarantees: in particular, stable forward propagation, robustness to input perturbations, which lead to improved model performance. Empirical evaluation obtained for various transformer architectures (encoders and decoders) on the small classification or language generation tasks shows superiority of the introduced approach in comparison to original transformer.

**Questions:**

My questions to the authors are as follows:

1. I understand that in this work you focused mainly on training the models on toy examples to demonstrate the applicability of the proposed approach. However, even for such “toy” tasks it would be useful to provide a more qualitative evaluation of the results. Would you consider evaluating the texts generated by the original GPT models (trained on different datasets) with established LLM benchmarks or an “LLM‑as‑a‑judge” framework? For instance, for the Shakespeare‑based nanoGPT, you could adopt the evaluation protocol introduced in TinyStories: How Small Can Language Models Be and Still Speak Coherent English?, where ChatGPT was used to assess the grammar, logic, and consistency of the generated stories. For the larger GPT‑2 models trained on general‑domain data, could you also report results on simple benchmarks such as GLUE?

2. Also I have a question regarding Figure 2, there, on the graph, we can observe an abrupt drop in training loss, and jump in training accuracy, while test loss starts coming upward on the 150 epoch. Is it connected to scheduler used during training. Can you, please, elaborate on this more?

3. Also the computational overhead is not discussed, do the OT-regularization introduce some memory or speed overhead for the final model?

**Ethical Concerns:**

["NO or VERY MINOR ethics concerns only"]

**Final Justification:**

I would keep my score. I believe that this work presents a novel more theoretical research, designed in a way, so that it can be applied to any Transformer-based model. The empirical evaluation is a little bit limited to small models, however, I don't see it as a disadvantage of the work, so, I believe it can be accepted for NeurIPS, and it will be a valuable and interesting paper for the community.

**Limitations:**

Yes

**Quality:**

3

**Strengths And Weaknesses:**

The main strengths of the paper are as follows:

1. A novel, clearly explained theoretical framework that allows to use optimal transport for training more robust Transformers-based models. The authors provide theoretical explanation for their method that seems clear and reasonable proving the stable forward propagation for the transformer-based models.
2. The simplicity of integration. While the method is quite novel, its plug-and-play design allows to use it without significant modifications to the original Transformers that makes it a good candidate for further development in the research groups.
3. Extensive experimental evaluation on different multimodal and language tasks and comparison with original transformers makes the theoretical research more solid. We can observe that the OT-Transformers trained in the same manner as the original ones outperforms the latter based on the accuracy, or the test loss.

The main weaknesses of the paper are as follows:
1. While the empirical evaluation is extensive, its plug-and-play design is still limited to the very simple tasks, like MNIST or binary classification, language generation is evaluated only based on the test loss that is usually not a completely fair point for benchmarking the language models. While the loss can be low, the generations themselves can be of different quality, for example, the model can loop into repetitions, and we are not able to notice it based on the pure loss.
2. In appendix G, the author notice the sensitivity of their model to the regularization parameter, however, it is noted only shallow, how this regularization term was adjusted (empirical study only)? Why this term turned out to be so important for the training stability is not properly discussed.

---

> ### Author Rebuttal · Authors · 2025-07-30
>
> We sincerely thank the reviewer for their time, effort, and constructive feedback. They have helped us improve and strengthen our paper. We are encouraged that the reviewer acknowledged the novelty, theoretical significance, and simplicity of implementation of our work. We address the reviewer's concerns in bullet points as follows. We also conducted additional experiments to address the comments of the reviewers.
> ___
> **Concern 1: Plug-and-play design is limited to simple tasks**
>
> **Response 1:** By plug-and-play, we mean **our model can be applied to any Transformer models** with only slight code modification. In particular, our model reuses existing Transformer architectures to parametrize a dynamical system (see Figure 1 and Equation 7).
>
> **Our approach is not limited to simple tasks and can be applied for any applications.** Our experiments cover problems with different scales, architectures, and applications. We specifically refer the reviewer to the GPT2 experiments in Appendix G, which shows that our model can handle realistic and hard tasks.
>
> Importantly, the plug-and-play nature, versatility, ease of adoption, and theoretically grounded performance improvement represent key advantages of our method.
> ___
> **Concern 2: Experiment GPT with LLM benchmarks**
>
> **Response 2:** We thank the reviewer for the helpful suggestion, we have added additional evaluation results for both the NanoGPT (shakespeare character-level) and GPT2 (OpenWebText) experiments using common LLM benchmarks including Perplexity, Bleu, Rouge1, Rouge2, RougeL, BertP, BertR, BertF1.  For all metrics, our model consistently outperforms the baseline, demonstrating the strength of our proposed approach. We will also add these results to the camera-ready version of the paper.
>
> **GPT2 (OpenWebText) experiment**
> | Model    | Test Loss   | Perplexity | BLEU (%)  | ROUGE-1 (%) | ROUGE-2 (%) | ROUGE-L (%) | BERT-P (%) | BERT-R (%) | BERT-F1 (%) |
> | -------- | ---------- | ---------- | --------- | ----------- | ----------- | ----------- | ---------- | ---------- | ----------- |
> | Baseline | 3.19     | 26.09      | 12.22     | 57.30       | 17.07       | 35.17       | 80.77      | 83.31      | 82.02       |
> | Ours     | **2.91** | **19.56**  | **14.07** | **59.46**   | **18.99**   | **38.05**   | **81.91**  | **84.27**  | **83.07**   |
>
> **NanoGPT (Shakespeare character-level) experiment** (We omit the Rouge and Bleu scores here, as they are less meaningful for character-level experiments; see [2-3])
> | Model    | Test loss | Perplexity   | BERT-P (%) | BERT-R (%) | BERT-F1 (%) |
> | -------- | --------- | ------------ | ---------- | ---------- | ----------- |
> | Baseline | 2.67      | 17.26    | 74.4      | 81.1      | 77.6       |
> | Ours     | **1.44**  | **4.40** | **74.8**  | **81.2**  | **77.9**   |
> ___
> **Concern 3: Explain Figure 2's loss plunge**
>
> **Response 3:** We adopt this experiment from [1], where it is used as a benchmark task. We use the same set of hyperparameters as the original experiment. These include the number of epochs and learning rate scheduler. At the 150th epoch, the learning rate decreases from 1e-3 to 1e-4. The drop in training loss and corresponding jump in test accuracy coincide with this scheduled change.
>
> We emphasize that the hyperparameters and training settings were originally chosen for the baseline and were not specifically tailored by us. Nevertheless, in all our seven extensive experiments, our model consistently achieves superior results even under suboptimal conditions.
> ___
> **Concern 4: Does the regularizer introduce more memory or speed overhead**
>
> **Response 4:** We emphasize that the regularization adds negligible computational and memory cost. This is because the regularization term is computed alongside Equation 7; see the discussion in Section 3.
> ___
> **Concern 5: Empirical sensitivity study of regularization term**
>
> **Response 5:** We observe that the model performance is stable with respect to different choices of $\lambda$ and other hyperparameters. Tuning of hyperparameters is generally not required; even if the optimal $\lambda$ and number of steps ($=\frac{T}{\Delta t}$) are not used, our model still outperforms the baseline.
>
> We conduct additional NanoGPT experiments with different $\lambda$'s and numbers of time steps ($=\frac{T}{\Delta t}$) for sensitivity study. The experiments are done over 3 random trials, and we report the average test loss and its standard deviation. We see that our model consistently outperforms the baseline across over two orders of magnitude of $\lambda$. Our model performance is also very stable with respect to the number of time steps. These additional results will be added to the camera-ready version.
>
> | \$\lambda\$ | 0.01         | 0.05         | 0.1          | 0.5          | 1                | 5            |Baseline            |
> | ----------- | ------------ | ------------ | ------------ | ------------ | ---------------- | ------------ | ------------ |
> | Test loss   | 2.68 ± 0.019 | 2.29 ± 0.017 | 2.05 ± 0.009 | 1.52 ± 0.002 | **1.44 ± 0.004** | 1.48 ± 0.003 | 2.68 ± 0.006 |
>
> | Time step (\$T/\Delta t\$) | 1            | 5                | 10           | 15           | 20           |Baseline            |
> | -------------------------- | ------------ | ---------------- | ------------ | ------------ | ------------ | ------------ |
> | Test loss                  | 1.46 ± 0.003 | **1.43 ± 0.002** | 1.44 ± 0.004 | 1.46 ± 0.002 | 1.49 ± 0.008 | 2.68 ± 0.006 |
> ___
> **Concern 6: Why this regularization term turned out to be so important for the training stability**
>
> **Response 6:** Indeed, this is one of the key points in our paper, and we have discussed this after Theorems 1 and 2 in Section 4, and full mathematical derivations are given in Appendices D2-D6. We here present a short version of our theory. We hope the additional discussion below helps clarify any confusion.
>
> We prove that for standard Transformer and when $\lambda=0$, the training problem is ill-posed in general. Specifically, there are infinitely many optimal solutions (trained Transformers) to the training problem. These include trained models with highly irregular (e.g., exploding) hidden states. This can explain why the model weights and gradients can explode during Transformer training especially when the model is deep; see the point cloud, sentiment analysis, NanoGPT, and GPT2 experiments in Appendix G.
>
> On the other hand, we prove that when the OT-Transformer with $\lambda>0$ is used, the optimal solution (trained Transformer) is unique. More importantly, we prove that the unique trained Transformer has highly regular hidden states; see Appendices D4-D6. Since **the OT regularization always leads to this unique and highly regular trained model, this can effectively prevent model weights and gradients from exploding and thus stabilize training.** These can explain the GPT-2 learning curves in Figure 8, where the Transformer architecture is deep and more prone to training instability, and our model's training and test curves are much less oscillatory than the baseline's.
> ___
> [1] Sander, M. E., Ablin, P., Blondel, M., & Peyré, G. (2022, May). Sinkformers: Transformers with doubly stochastic attention. In International Conference on Artificial Intelligence and Statistics (pp. 3515-3530). PMLR.
>
> [2] Gehrmann, S., Adewumi, T., Aggarwal, K., Ammanamanchi, P. S., Aremu, A., Bosselut, A., ... & Zhou, J. (2021, August). The GEM Benchmark: Natural Language Generation, its Evaluation and Metrics. In Proceedings of the 1st Workshop on Natural Language Generation, Evaluation, and Metrics (GEM 2021) (pp. 96-120).
>
> [3] Denoual, E., & Lepage, Y. (2005). BLEU in characters: towards automatic MT evaluation in languages without word delimiters. In Companion Volume to the Proceedings of Conference including Posters/Demos and tutorial abstracts.

---

> > ### Comment · Reviewer_fc21 · 2025-08-07
> > **Official Comment by reviewer fc21**
> >
> > Thank you for the thorough and detailed response. The additional experiments and clarifications directly address the main issues I raised.
> >
> > Specifically, the broader evaluation of the LLMs (NanoGPT and GPT‑2) demonstrates that the regularized models improve not only cross‑entropy loss but also generation quality on realistic language tasks.
> >
> > Thank you also for clarifying that the hyperparameters were taken from the original experimental setup to ensure a fair comparison. It will be interesting to analyze which set of hyperparameters is most suitable for your approach.

---

> > > ### Author Response · Authors · 2025-08-07
> > > **Additional Response**
> > >
> > > Thank you for your response. We are glad that our rebuttal has directly addressed your concerns. We address the additional comment in the following.
> > >
> > > ___
> > >
> > > **Concern 7: It will be interesting to analyze which set of hyperparameters is most suitable for your approach.**
> > >
> > > **Response 7:** Since we did not perform tuning on training hyperparameters, doing so will almost certainly enhance our model's performance, resulting in an even greater performance gain over the baseline models (while reducing model size).
> > >
> > > However, hyperparameter tuning is neither required nor the focus of our work. This is because our model is designed to be plug-and-play and theoretically guaranteed to be robust. Our model outperforms the baseline across a wide range of hyperparameters, as reported in the sensitivity analysis of our rebuttal.

---

### Note · Authors · 2025-08-12

We would like to sincerely thank the AC and reviewers again for the time and care you have put into reading our work and sharing your feedback. The valuable discussion has helped us better explain our ideas and make meaningful improvements to the paper. We believe we have comprehensively addressed all questions and concerns through numerous additional simulations and in-depth discussions with the reviewers who engaged during the rebuttal period.

In the following, we summarize our work and the discussion in bullet points.

- **Our theory is a key novelty and contribution** in the study of Transformers, setting us apart from prior work. It provides theoretical guarantees on generalization and robustness, which are empirically validated by – and in turn support – our initial and added numerical experiments.
- **Our approach is designed to be plug-and-play.** It can be applied to any Transformer models and any applications with only slight modification to the code. It leads to theoretically guaranteed performance improvement over the baseline model. This is achieved while reducing model size, even without hyperparameter tuning. The performance gain can be further enhanced through hyperparameter tuning.
- We **conduct extensive experiments across diverse applications and challenging conditions**, demonstrating the effectiveness, generalizability, robustness and scalability of our proposed approach. Specifically, our experiments are broad in that they cover diverse architectures, tasks, data types and scales. They are also deep, including noisy settings to evaluate robustness and tests for out-of-distribution generalization.
- We emphasize again that our method consistently improves performance on both training and test data, indicating benefits beyond merely reducing overfitting.
- Like other continuous-time models, our approach may incur additional computational cost. However, this overhead is compensated by the performance gain, proven generalization & robustness  and reduced parameter count. Moreover, thanks to the theoretically-guaranteed and empirically demonstrated regularity of the hidden dynamics, our method achieves superior results to other continuous-time models in both performance and computational efficiency.

We thank you again for your engagement and your suggestions. A more detailed pseudocode in algorithm format, as well as the additional numerical experiments and clarifications will be updated to the manuscript to ensure better readability.

---

### Decision · Program_Chairs · 2025-09-17

**Decision:**

Accept (poster)

**Comment:**

This paper proposes a novel plug-and-play model called OT-Transformer, which formulates Transformers in a continuous-time setting and adds an optimal transport (OT) regularization into the training loss. With a large enough regularization parameter $\lambda > TL^2$, the model comes with a property dubbed stable forward propagation (Theorem 2), which states that if two input–target output pairs are similar to each other, then the output of the optimally trained model will also be similar; the property is shown to enhance generalization and robustness (Theorem 3). The proposed model is tested with seven experiments across different modalities and tasks, exhibiting consistent improvements over baselines.

Most reviewers appreciated the novelty of the proposed framework. Reviewers ZUmk and SwmS raised a concern that the robustness and generalization implications of the stable forward propagation property were not properly tested in the paper. With the authors’ additional experiments during the rebuttal phase, I believe that the issue is addressed. After a careful review of the entire discussion, I found that most other questions and concerns were well-addressed. Some reviewer commented that the evaluations were done only on relatively small models, but they also added that this is not viewed as a disadvantage; I also agree with this view, given the diversity of experiments presented.

Overall, the paper presents a novel plug-and-play framework that can be potentially impactful. I recommend acceptance. As noted in the Author Final Remarks, please include the additional LLM benchmark results and robustness results, which will further strengthen the paper. Please also include the runtime analysis (in the response to Reviewer xcGF), as this will better reflect the time–performance trade-off.